# Predicting the Production and Depletion of Rare Earth Elements and Their Influence on Energy Sector Sustainability through the Utilization of Multilevel Linear Prediction Mixed-Effects Models with R Software

Hamza El Azhari [1,*], El Khalil Cherif [2,3,*], Rachid El Halimi [4], El Mustapha Azzirgue [1], Yassine Ou Larbi [4], Franco Coren [2] and Farida Salmoun [1]

1 Laboratory of Physical Chemistry of Materials, Natural Substances and Environment, Chemistry Department, Faculty of Sciences and Technology Tanger, Abdelmalek Essaâdi University, Tangier 90090, Morocco; elmustapha.azzirgue1@etu.uae.ac.ma (E.M.A.); f.salmoune@uae.ac.ma (F.S.)
2 National Institute of Oceanography and Applied Geophysics (OGS), Centre for Management of Maritime Infrastructure (CGN), Borgo Grotta Gigante 42/C, Sgonico, 34010 Trieste, Italy; fcoren@ogs.it
3 MARETEC—Marine, Environment and Technology Center, Instituto Superior Tecnico, Universidade de Lisboa, Av. Rovisco Pais 1, 1049-001 Lisboa, Portugal
4 Department of Mathematics and Statistics, Faculty of Sciences and Techniques of Tangier (FST), Abdelmlek Essaadi University (UAE), Tétouan 93000, Morocco; r.elhalimi@uae.ac.ma (R.E.H.); yassineoularbi2@gmail.com (Y.O.L.)
* Correspondence: hamza.elazhari@etu.uae.ac.ma (H.E.A.); c.elkhalil@uae.ac.ma (E.K.C.); Tel.: +212-635324751 (H.E.A.)

**Abstract:** For many years, rare earth elements (REEs) have been part of a wide range of applications (from cell phones and batteries to electric vehicles and wind turbines) needed for daily life all over the world. Moreover, they are often declared to be part of "green technology". Therefore, the data obtained from the United States Geological Survey (USGS) on the reserve and production of rare earth elements underwent treatment using the multivariate imputation by chained equations (MICE) algorithm to recover missing data. Initially, a simple linear regression model was chosen, which only considered fixed effects ($\beta$) and ignored random effects ($U_i$). However, recognizing the importance of accounting for random effects, the study subsequently employed the multilevel Linear Mixed-Effects (LME) model. This model allows for the simultaneous estimation of both fixed effects and random effects, followed by the estimation of variance parameters ($\gamma$, $\rho$, and $\sigma^2$). The study demonstrated that the adjusted values closely align with the actual values, as indicated by the p-values being less than 0.05. Moreover, this model effectively captures the sample's error, fixed, and random components. Also, in this range, the findings indicated two standard deviation measurements for fixed and random effects, along with a variance measurement, which exhibits significant predictive capabilities. Furthermore, within this timeframe, the study provided predictions for world reserves of rare earth elements in various countries until 2053, as well as world production forecasts through 2051. Notably, China is expected to maintain its dominant position in both reserve and production, with an estimated production volume of 101,985.246 tons, followed by the USA with a production volume of 15,850.642 tons. This study also highlights the periodic nature of production, with a specific scale, as well as periodicity in reserve. These insights can be utilized to define and quantify sustainability and to mitigate environmental hazards associated with the use of rare earth materials in the energy industry. Additionally, they can aid in making informed decisions regarding at-risk rare earth reserves, considering potential future trends in electric vehicle (EV) production up to the year 2050.

**Keywords:** rare earth elements; multilevel linear mixed effects (LME); linear regression; USGS; MICE; global economic countries; quantify sustainability; electric vehicle (EV)

## 1. Introduction

The global embrace of modern technologies (MTs) has profoundly impacted various domains, spanning medicine, machine learning, computer science, telecommunications, and environmental and natural sciences. Within this context, REEs have gained significant importance due to their unique physical, chemical, catalytic, electrical, and magnetic properties [1,2]. As indispensable resources, REEs play crucial roles in diverse technological applications, including intelligent and sustainable technologies and industrial raw materials. Comprising 17 elements such as lanthanum, cerium, praseodymium, and yttrium [1], REEs are not as scarce as their name implies, Abundant in the Earth's crust, their prevalence exceeds that of silver (Ag) and mercury (Hg) [3]. The initial discovery dates back to 1788, when Johan Gadolin isolated an oxide named "yttria," later identified as a mixture of various rare earth oxides.

REEs are categorized into light (LREEs) and heavy (HREEs) groups based on atomic numbers [4]. The LREEs are eight elements with atomic numbers ranging from 57 to 63 (La to Eu) [5]. The HREEs group consists of seven elements with atomic numbers ranging from 64 to 71 (Gd to Lu) [6]. The classification varies among scholars and organizations, with perspectives on whether scandium (Sc) and yttrium (Y) should be included. Balaram [7] mentions that the International Union of Pure and Applied Chemistry (IUPAC) classifies the elements from La to Eu as light REEs, while Gd to Lu, along with Y, are considered heavy REEs. The classification of Gd also varies between the two groups, as indicated by the USGS [8]. Additionally, scandium (Sc) is often treated separately as an element and not considered part of the REEs [9]. The distribution of REEs in the Earth's crust follows the Oddo-Harkins rule [10,11], with even-atomic-numbered elements being more abundant. Found in various mineral deposits, REEs typically exist as trivalent cations in carbonates, oxides, phosphates, halides, and silicates [12,13]. However, the concentrations in these deposits are relatively low, ranging from 10 to a few hundred parts per million (ppm) by weight, making extraction challenging [14].

Cerium was the first REE to be isolated [15] in the 19th century, followed by the extraction of all naturally occurring REEs, concluding with the discovery of promethium in 1947, encapsulating over 150 years of exploration [16]. At the beginning of the 20th century, global annual REE production and consumption were below 5000 tons, with limited integration into daily life until the 1960s [17]. the 1960s saw a transformative in production and consumption, exceeding 75 thousand tons [18], driven by applications in everyday items and technological advancements. By the 21st century, REEs became integral to over 200 high-tech products, and played a crucial role in emerging clean energy technologies [19].

As the global shift towards clean energy and electrification gained momentum, rare earth resources became crucial, leading to increased competition among nations [20,21]. In the last decade, China emerged as the dominant supplier, producing over 85% of the world's rare earth oxide. REEs are considered critical raw materials, alongside other strategic minerals [22,23] which are discovered within the Earth's crust, forming mixtures of various REEs and some non-metals. Cerium stands out as an exception, being relatively more abundant in the Earth's crust [24]. Generally, REEs coexist with other strategic mineral resources like cobalt, lithium, tellurium, and nickel [25].

Extensive research has identified approximately 200 minerals containing REEs, but only a few, such as bastnasite and monazite, are economically feasible for mining [13]. Bastnasite (70–75% REO), (55–60% REO), and xenotime (55–60% REO) contribute to about 95% of the world's REE reserves [26].

The presence of missing data is a common issue encountered in real-world datasets [27]. Addressing missing data is handled through MICE under certain assumptions, ensuring a comprehensive analysis [28]. The missing values are imputed based on either the order of the data columns or a pre-determined sequence [29].

Future projections of rare earth production and reserves, both regionally and internationally, are crucial [30], as numerous studies advocate for the use of mathematical models



as effective and versatile tools for analyzing forecasts, evaluating scenarios, and monitoring associated risks related to the depletion of these mineral resources. While notable examples of these models used are Gaussian, Gompertz, Weibull, Rayleigh, and log-normal, all of which have been employed in the fossil fuel sector, the results have been unsatisfactory, necessitating more flexible models [31]. Scholars [32] have introduced models such as the Hubbert, Logistic, Generalized Weng, and the Richards model, which is a comprehensive model incorporating Logistic and Gompertz curves. In the same context, LME has been extensively studied and applied in various disciplines like hydrology, meteorology, and biology [33,34]. However, classical diagnostics face challenges in LME models, with disruptions to asymptotic results, confounding factors, and visible patterns in residual plots [35].

LME models incorporate both fixed and random effects linearly, effectively considering the correlation between observations within the same sample group [36]. Model monitoring is crucial for valid inference [37], involving the examination of model assumptions and assessing how well the model captures data characteristics. Test statistics and *p*-values can be used to evaluate evidence strength, although these methods only indicate potential model issues [38].

The increasing importance of rare earth industries has generated widespread interest in examining global trends in rare earth supply, particularly in light of the growing demand for rare earths in renewable energy technologies. This interest is underscored by re-source limitations, as highlighted in various studies [39,40]. However, the existing literature indicates that prior research has predominantly focused on estimating rare earth supply using experiential judgment on current and anticipated production capacities, neglecting quantitative modeling approaches. Moreover, these studies have overlooked future rare earth consumption, failing to account for potential demand increases from users with limited resources [41]. As a result, the findings primarily offer short-term production estimations, often confined to a single country, without providing insights into the long-term outlook.

This study aims to forecast the future production of rare earths globally by 2051 and mitigate the risk of geological reserve depletion until 2053, relying on data from the USGS [42]. The objectives are as follows:

1-  Demonstrate the efficacy of MLE models for the precise evaluation of future production and reserves.
2-  Identify periods characterizing the highest consumption and production of rare earths in the upcoming years, offering a decision-support system for stakeholders to formulate strategies and a preventive plan to manage and avoid depletion of the geological reserve.
3-  Emphasize the importance of employing MLE for accurate predictions of worldwide reserves and production of REE at global and country-specific levels.

## 2. Materials and Methods

### 2.1. Literature Review and Rare Earth Data Sources

The history of REEs traces back to the late 19th century when Swedish chemist Carl Axel Arrhenius identified a new mineral containing an unknown element, Cerium. Over subsequent decades, more REEs such as Lanthanum, Praseodymium, and Neodymium were discovered [43].

In the early 20th century, REEs found applications in gas mantle production and alloy creation for aviation. The real surge in demand occurred in the 1960s and 1970s, driven by applications like color televisions utilizing europium for the red hue, lasers, and magnets [44]. Initially, South Africa, India, and Brazil were major producers, but the landscape changed in the 1960s. The Mountain Pass deposit in California, discovered in 1949, made Australia and the United States key players. The U.S. dominated production from 1965 to 1995 [45,46]. However, China began to strengthen its industrial power and enter global trade in the late 1970s. Although it started to develop the technology of

extraction, separation, smelting, and further processing of RE minerals in the late 1950s, it significantly increased production in the early 1980s, achieving an annual output growth of about 40%, totaling around 150 tons [24].

By the late 1990s, China had established control over the REE market. This led to a significant drop in REE prices, causing various enterprises [47], including the American Mountain Pass Mine, to cease operations [48]. China's dominance increased, eventually reaching 95% of global production as it acquired stakes in REE firms worldwide, resulting in a "Rare Earth Crisis" in 2010. A dispute between China and Japan led to China imposing restrictions on REE exports, causing prices to surge to 500% in some cases [49]. This incident prompted several nations to reconsider their REE possibilities to reduce their reliance on China. Australia initiated REE mining operations in 2011, and the United States, Canada and, subsequently, Brazil, Malaysia, Russia, Thailand, and Vietnam followed. Although other countries' contributions have been growing, China maintains its dominance in the global REE market. In 2022, global REE production reached an estimated 300,000 tons, doubling the output from 2016. China accounted for nearly 70% of the total supply (210,000 tons), followed by the United States (14%) and Australia (4%) [42]. The estimated global reserves of rare earth elements are around 120 million tons (earth oxide equivalent), with China holding approximately 37% of the total, according to the USGS [8].

The extensive utilization of REEs in diesel fuel additives due to their catalytic properties is well-documented in numerous studies [24], with lanthanum being the primary element in varying concentrations between 0.001% and 0.05%, resulting in an annual demand of around 200 tons [50]. While the growing prominence of hydrogen fuel cell vehicles may influence global REE consumption, the predominant application lies in hybrid and electric vehicles [51], particularly in the engine battery. These vehicles, classified into five main types, represent a significant sector for REE utilization, as outlined in Table 1 [52]:

**Table 1.** The categorization of electric vehicles.

| | |
|---|---|
| Battery Electric Vehicles (BEVs) [53] | These are 100% electric vehicles that rely solely on electric power. BEVs use large battery packs to provide a driving range ranging from 160 to 500 km with a single charge. Examples include the Nissan Leaf, with a 62 kWh battery offering a 360 km range. |
| Plug-In Hybrid Electric Vehicles (PHEVs) [54] | PHEVs combine a conventional internal combustion engine with an electric engine that can be charged from an external source. They can significantly reduce fuel consumption by utilizing electricity from the grid. The Mitsubishi Outlander PHEV, with a 12 kWh battery, can drive around 50 km on electric power. |
| Hybrid Electric Vehicles (HEVs) [55,56] | HEVs also combine an internal combustion engine with an electric engine, but they cannot be charged externally. The battery is charged through the vehicle's combustion engine and regenerative braking. The Toyota Prius, as a hybrid model, offers a 1.3 kWh battery with an electric range of up to 25 km. |
| Fuel Cell Electric Vehicles (FCEVs) [57] | FCEVs use compressed hydrogen and oxygen to power an electric engine, emitting only water as a byproduct. However, most hydrogen is derived from natural gas. The Hyundai Nexo FCEV can travel up to 650 km without refueling. |
| Extended-range EVs (ER-EVs) [58] | Similar to BEVs, ER-EVs have large batteries for electric driving. Additionally, they are equipped with a supplementary combustion engine that charges the batteries when needed. The BMW i3, for instance, offers a 42.2 kWh battery providing a 260 km electric range, with an extra 130 km available in extended-range mode. |

For instance, a popular hybrid vehicle's battery pack contains 10–15 kg of lanthanum [59]. In 2010, approximately 740,000 new hybrid vehicles were registered worldwide, leading to an estimated demand of 9300 tons of REEs per year, assuming an average of 12.5 kg of REEs per vehicle. Additionally, an electric vehicle typically requires around 200 g of neodymium for the motor's permanent magnets [60]. For the high scenario, hybrid electric vehicles dominate the electric vehicle market, accounting for approximately 70% of sales in 2017 [55].

REEs play a vital role not only in hybrid and electric vehicles but also in catalytic converters (CATCONS), due to cerium's ability to catalyze the conversion of nitrous oxides

into elemental nitrogen. CATCONS are present in approximately 85% of cars and light trucks produced globally as of 2013 [61]. The average amounts of cerium present in

CATCONS vary for specific vehicle categories, which can be classified into 3 groups (Figure 1):

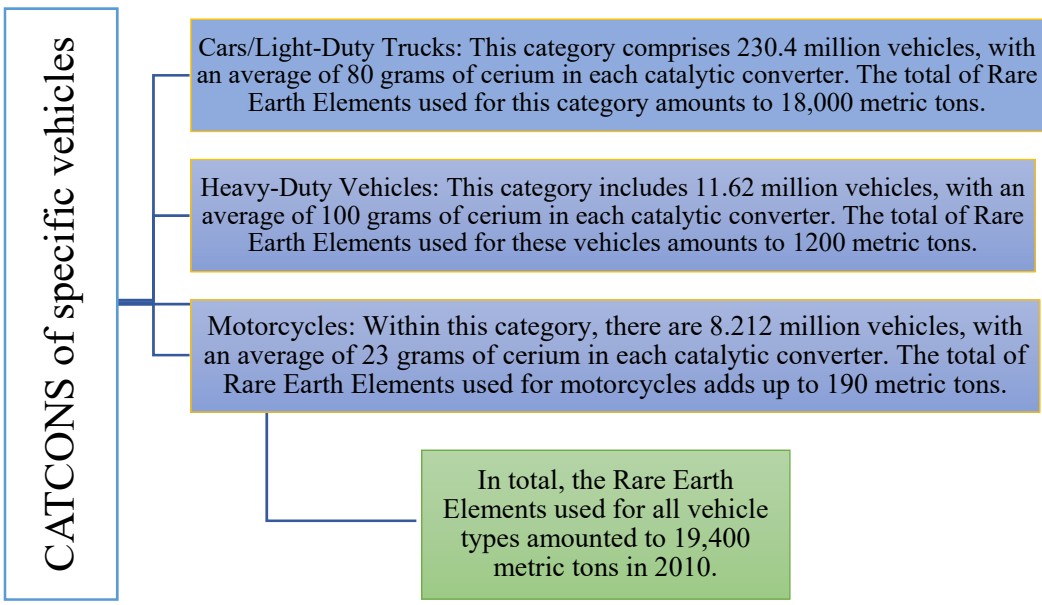

**Figure 1.** CATCONS of specific vehicles.

The 2012 sedan model exemplifies the inclusion of REEs in vehicles. This particular vehicle was found to contain around 0.44 kg of REEs, with 0.4 kg integral to the vehicle's fundamental functions including safety components and the basic radio/speaker system, and 0.04 kg designated for additional features such as the navigation system, DVD player, and electrically controlled seats [62]. However, hybrid vehicles, specifically those equipped with a nickel–metal hydride battery, demonstrate a higher REE of over 3.5 kg [63]. On the other hand, a hybrid electric vehicle equipped with a lithium-ion battery contains around 1 kg of REE, falling within the limit estimates set by Molycorp of 10 kg of rare earth per electric vehicle [50].

*2.2. Data Processing*

The global production and reserve data (Figures 2 and 3) for REEs were sourced from historical datasets provided by the USGS. These datasets cover the periods 1994–2023 for reserves and 1994–2022 for production. The objective was to analyze trends in reserves and production across different countries and assess the impact of production on reserves. The study involves two main periods: historical data analysis from 1994 to 2023 for reserves and from 1994 to 2022 for production. Future projections were made for the periods 2024–2053 for reserves and 2023–2051 for production. The growth rate of the estimated primary production of REEs was determined based on the anticipated future demand projected for the year 2030 [40]. The analysis employed two modeling approaches: a regression linear module and an LME module. This comparison aimed to evaluate the effectiveness and performance of the two models.

Figure 2 illustrates the historical shifts in global REE production, with the United States initially leading before China surpassed it in the mid-1990s, becoming the largest producer. China's rapid production increase and cost advantages resulted in a maximum annual production of approximately 43,000 tons, while other countries reached a minimum of 80 tons. Figure 3 showcases a sinusoidal pattern in the time series, indicating cyclical variation in the maximum annual reserve of rare earth elements. Notably, China's reserve can reach up

to 44,000,000 tons. The data analysis revealed that the time series variable lacks stationarity, indicating inconsistent mean and variance over time with observable fluctuations.

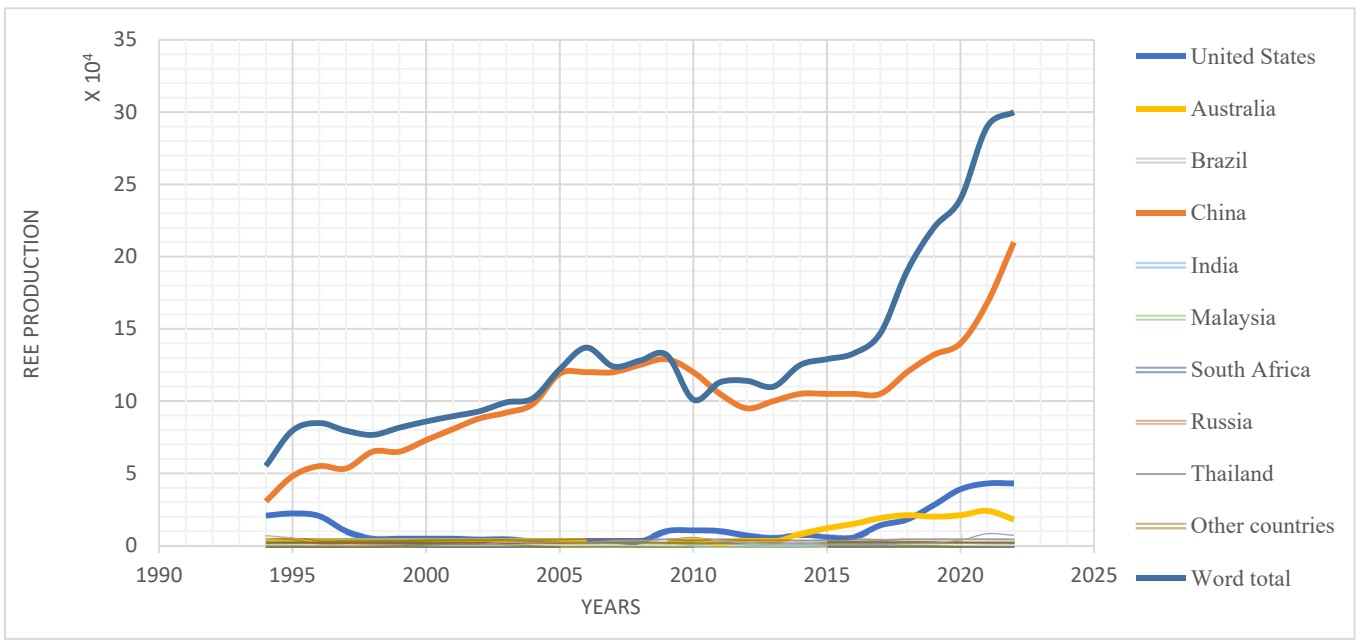

**Figure 2.** A visual representation of the evolution of REE production over time from 1994 to 2022.

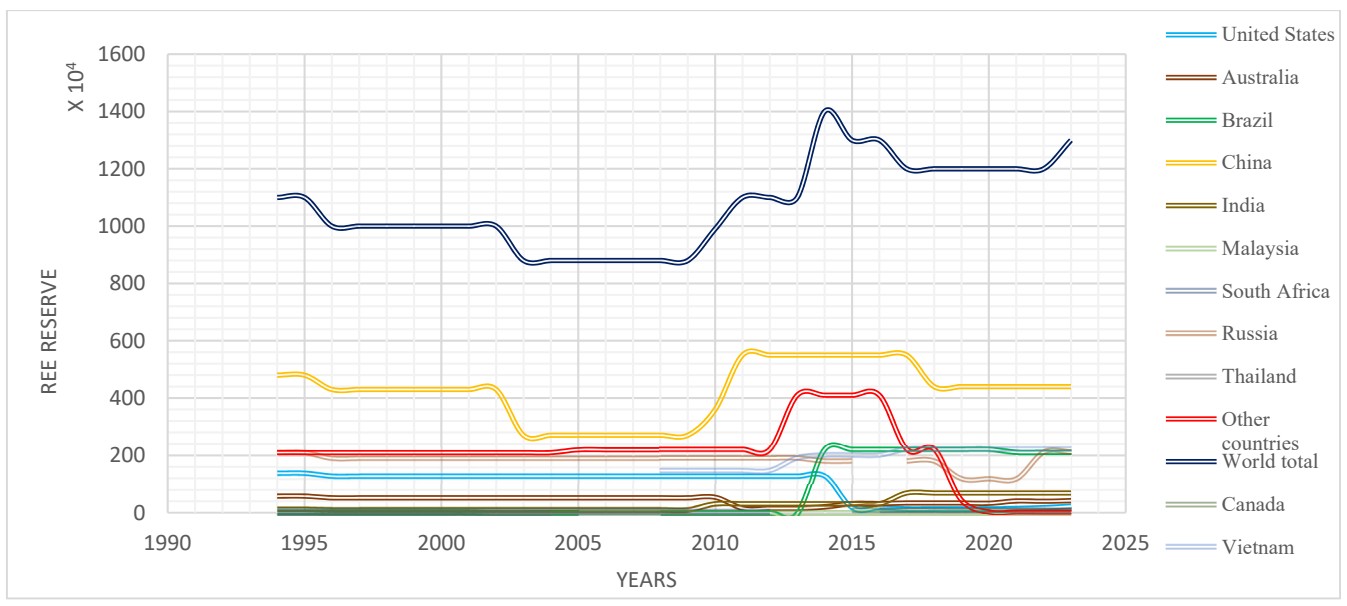

**Figure 3.** A visual representation of the evolution of REE reserves over the time span of 1994–2023.

### 2.3. Statistics Method

### 2.3.1. The Gap-Filling Method (MICE Algorithm)

In order to fill in the incomplete data for the different countries, we applied an imputation method: the multivariate imputation method by chained equations (MICE). This method was chosen because of its flexibility and use in various domains [64]. Also, the performance of this method was evaluated by statistical tests of performance evaluation.

The MICE approach was developed by B. Rubin in 1987 and is also called the "multiple sequential regression imputation". It is a semi-parametric imputation approach that allows the suggestion of a model closer to the nature of the data [65]. Its principle is based on the

imputation of incomplete variables [66–68]. On the one hand, in a successive variable-by-variable manner, it assigns a specific regression model to each variable, On the other hand, it uses an iterative way to obtain convergence according to the MICE [69], as described below [70,71]:

- Specify an imputation model $P(Y_j^{mis}|Y_j^{abs}; R_{-j}, R)$ for the variable $Y_- - 1$ with j = 1, ..., p.
- For each j, fill in the starting charges $\dot{Y}_j^{mis}$ by random draws among $Y_j^{abs}$.
- Repeat the procedure for t = 1..., T.
- Repeat the procedure for j = 1..., p.
- Define $\dot{Y}_1^1 = (\dot{Y}_1^t; ....; \dot{Y}_{j-1}^t, \dot{Y}_{j=1}^{t+1}, ..., \dot{Y}_p^{t-1})$ as currently complete data, except for $Y_j$.
- Pull $\dot{\varnothing}_1^t \sim P(\varnothing_j^t|Y_j^{abs}, \dot{y}_{-1}; R)$.
- Draw up imputations $\dot{Y}_j^1 \sim P(Y_j^{mis}|Y_j^{abs}, \dot{Y}_{-1}^t, R, \dot{\varnothing}_1^t)$.
- End of repetition j.
- End of repetition t.
- The estimated coefficients of the imputation model for the i-th variable"$\hat{\beta}_i$".
- The estimated covariance matrix of the imputation model residuals "$\hat{\Sigma}$".
- The vector of imputation model residuals "$\epsilon$".
- The number of imputations "R".
- The index for the current imputation "m".
- The number of iterations within each imputation "Q".
- The convergence criterion for each imputation "$\rho$".

Here, P is the probability that simulates the Bayesian distribution (a) posterior of the missing variable; Y is a matrix of dimension n × p containing the realizations of p variables for experimental units; and R is the vector of response indicators whose components $r_{ij}(i = 1...; n = 1$ and $j = 1, ..., p)$ is 1 if Y is observed and 0 otherwise. The missing parts of Y are noted $Y_j^{abs}$ and $Y_j^{mis}$, respectively. $\dot{Y}$ represents the values $n_0$ of the imputed vector in Y, $Y_{-j}$ all the columns of Y except for $Y_j$, and $\dot{\phi}$ represent the unknown parameters of the imputation model. In practical terms, this multiple imputation method is done in several steps. First, the choice of a regression model for the variable under study. Second, assigning a random value to each missing data value with a random value from the observed data, and this in an iterative way. Finally, the estimation of the imputed values according to the regression coefficient is estimated on each data set [72].

### 2.3.2. Theoretical Modeling

Similar to fossil fuels, rare earth elements are finite resources that can be depleted [73]. It is crucial to consider the availability of these exhaustible resources as it imposes limits on their long-term production. Consequently, it is highly likely that the long-term production trajectory for these resources initially increases [74]. Several models have been used to analyze the long-term production of minerals, such as the bell-shaped pattern commonly used to predict fossil fuel production and applied to understand the production behavior of various non-fossil fuel minerals [17]. For instance, Vikström [75] utilized logistic and Gompertz models to predict global lithium production, while Walan [76] employed the same models to study phosphate rock production based on reserve estimates in Morocco. Wang [77] used the generalized Weng model to forecast China's rare earth production. Gann [78] conducted a study using the Hubbert model to examine rare earth element (REE) production. In a separate study, Uliyanin employed the Hubbert model to analyze the depletion of uranium resources [79].

### 2.3.3. Application of Linear Mixed-Effects Models

A mixed model is a powerful statistical tool that examines the relationships between observed responses and explanatory covariates while accounting for two types of varia-

tions [80]. By incorporating fixed and random effects, it captures systematic influences and inherent data variability [81]. It recognizes the influence of both covariates and unobserved factors on the observed response. This comprehensive approach allows for the analysis of complex datasets, revealing insights into relationships and dynamics often overlooked by traditional statistical approaches. Practical examples of model construction using mixed-effects models can be found in [33,82]. The general linear mixed model assumes that the outcome vector, denoted as $Y_i$, consisting of $n_i$ outcomes for individual i, satisfies a certain condition (Equation (1))

$$Y_i = X_i\beta + Z_iU_i + \varepsilon_i \tag{1}$$

where i = 1, . . ., L.

In this model, $\beta$ represents a p-dimensional vector of unknown population-average regression coefficients, commonly known as fixed effects dimensions p. $X_i$ represents a known design matrix with dimensions $n_i \times p$, which connects the regression coefficients $\beta$ to the response variable $Y_i$.

The individual effects, denoted as $U_i$, are represented by a vector with dimensions $q \times 1$. These effects are assumed to follow a normal distribution, centered at 0, with a covariance matrix D. These random effects describe the deviation of the ith individual's evolution from the average evolution in the population. The design matrix $Z_i$, with dimensions $n_i \times q$, connects $U_i$ to $Y_i$. The residuals $\varepsilon_i$ are assumed to be independent and normally distributed N (0, $R_i$), with a mean 0 and a covariance matrix $R_i$. The dimension of $R_i$ depends only on I through its dimension $n_i$, while the unknown parameters in $R_i$ do not depend on i. It is worth noting that the random effects $U_i$ and the residuals $\varepsilon_i$ are assumed to be independent across different individuals and independent of each other for the same individual. This model can be further simplified when $R_i = \sigma^2 I$ and $D = \sigma_U^2 I$, where $\sigma^2$ and $\sigma_U^2$ are scalars, and I represents an identity matrix. This model, which includes two sources of random variation ($U_i$ and $\varepsilon_i$), is sometimes referred to as a hierarchical model [83,84] or a multilevel model [85].

The estimation of the mixed linear model (1) requires learning the unknown parameters: $\beta$, $\sigma_U^2$, and $\sigma^2$ from the available data. This step is essential in implementing the mixed linear model. The parameters of the mixed linear model, namely $\beta$, $\sigma_U^2$, and $\sigma^2$, can be estimated through likelihood maximization (ML) or a related method called restricted maximum likelihood estimate (REML) [86].

These parameters encompass the components of the variance, which describe the variability within the model [87]. Additionally, the model involves estimating the parameters associated with the fixed effects, which represent the relationships between the predictors and the response variable. Furthermore, the model includes the realizations of the random effects, which capture the individual-specific or group-specific variations. The process of implementing the mixed linear model entails estimating these different parameters through various statistical methods such as maximum likelihood estimation or restricted maximum likelihood estimation [88].

The maximum likelihood (ML) method is known as relatively strong, providing estimators with optimal asymptotic properties. It is widely used and has desirable asymptotic properties, and it can yield biased estimators for variance parameters [89]. However, The REML (restricted maximum likelihood) method, derived from maximum likelihood, specifically addresses the loss of degrees of freedom caused by estimating fixed parameters. It takes into account this issue to provide more reliable estimations [90]. This method is particularly useful in the context of linear mixed-effects (LME) models. Gomes [91] states that the formulation (1) for simple-level LME models can be extended to accommodate multiple levels of random effects.

The equation mentioned above represents the Laird–Ware [92] formulation for single-level LME models. It is expressed for the $n_i$ dimensional response vector Yi for the ith individual as (Equation (1)). This formulation can also be extended to accommodate multiple nested levels of random effects. For instance, in the case of two nested levels of random effects, the response vectors $Y_{ij}$ of length $n_{ij}$, where i = 1,. . ., L, and j = 1,. . ., $L_i$ (L

represents the number of individuals in the first level, while $L_i$ represents the number of second-level individuals within the first-level individual i), can be modeled in vector form as follows [93] (Equation (2))

$$Y_{ij} = X_{ij}\beta + Z_{(i)'_j}U_i + Z_{ij}U_{ij} + \varepsilon_{ij} \tag{2}$$

where i = 1 ..., L, and j = 1 ..., $L_i$.

The first-level random effects, denoted as $U_i$ with a length of $q_1$, are assumed to be independent across different individuals i. The corresponding model matrices, $Z_{(i)'_j}$ with dimensions $n_i \times q_1$, are used to capture these random effects. On the other hand, the second-level random effects, $U_{ij}$ with a length of $q_2$, are assumed to be independent for different i or j. The corresponding model matrices, $Z_{ij}$ with dimensions $n_i \times q_2$, are utilized to represent these random effects. Importantly, the second-level random effects are also independent of the first-level random effects.

The fixed-effects model matrices, denoted as $X_{ij}$, are of size $n_{ij} \times p$, and are used to incorporate the fixed effects. These matrices are specific to each combination of i and j, where i = 1, ..., L and j = 1, ..., $L_i$. The error terms $\varepsilon_{ij}$ are random variables with a dimensional of $n_{ij}$. They are assumed to be mutually independent for different i or j, and typically follow a Gaussian distribution, $\varepsilon_{ij} \sim N(0, \sigma^2 I)$.

Mixed-effects models have gained popularity, particularly for analyzing quantitative data that exhibit a multilevel structure. These models enable the calculation of descriptive statistics and predictive measures at different stages, considering the inherent hierarchy within the data. This hierarchical nature is taken into account as the data flows through subsequent steps of analysis [31]:

- Step 1: Choose a suitable mathematical function that can be fit to the data set;
- Step 2: Perform curve-fitting and introduce constraints to improve fit quality;
- Step 3: Extrapolate the fitted model to project future production trends.

*2.4. Model Comparisons (Applications to Rare Earth Data)*

2.4.1. Adapted Modeling for Reserve and Production

The paper presents "complete" data, which is described in the final section (Appendix A) and visually represented in Figures 2 and 3. This data provides information on the cyclical patterns of production across 11 different countries based on the economically significant nations (Refer to Table 1 to observe the Hierarchical Data Structure) for a span of 29 years (1994–2022), in addition to the cyclical patterns of the reserve across 13 different countries for a span of 30 years (1994–2023). Upon observing Figures 2 and 3, it becomes evident that there are significant differences between countries, as well as some indications of variations between years. Furthermore, the countries demonstrate variability in their production and reserve scores within each year.

In our analysis, we aim to model these differences, whether it involves some or all of the observed variations. However, at this stage, we initially disregard the data's grouping structure and assume a simplistic basic regression model. Let $Y_{ij}$ represent the response variable ("REE reserve or production") for county "I" in year "j". Assume that $Y_{ij}$ is connected to a sequential time variable $X_{ij}$ through a logarithmic function, $y_{ij} = \log(Y_{ij})$, where i = 1, 2, ..., n and j = 1, 2, ..., $n_i$. If the regression model's parameters remain constant across countries and different years (no heterogeneity), we arrive at a nonlinear regression model (Equation (3))

$$Y_{ij} = \beta_0 + \beta_1 \sin(w \times X_{ij} + T) + \varepsilon_{ij} \tag{3}$$

where

- $Y_{ij}$: Represents the response variable;
- w: Represents the frequency of the sinus wave, approximately 0.5;
- T: Represents the periodic pattern $\approx \frac{\pi}{2}$;
- $\beta_0$: Comprises the mean intercept;

- $\beta_1$ Represents the fixed effects for the population, and the common slope or growth rate;
- $E_{ij}$: Represents the independent error term where $\varepsilon_{ij} \sim N(0, \sigma^2)$;
- In the case where ($Y_{ij}$ = Production), N = 13, and $n_1 = n_2 = \ldots = n_{13} = 29$. Alternatively, if ($Y_{ij}$ = Reserve), N is equal to 11, and $n_1 = n_2 = \ldots = n_{13}$. The total number of observations is $N = \sum_{i=1}^{29} n_i = 319$.

It is evident that this model assumes a constant value for all parameters $\beta_0$, $\beta_1$, and $\sigma^2$ across countries. In order to evaluate the adequacy of the current model and explore possible modifications, graphical and numerical diagnostic tools are employed. The equation mentioned previously yields the following results: the estimated intercept ($\hat{\beta}_0$) is 7.902996 (*p*-value = 0), the estimated slope ($\hat{\beta}_1$) is $-0.244078$ (*p*-value = 0.1794), and the residual variance unexplained by the model (3) ($\hat{\sigma}^2 = 2.28606^2 = 5.22607$). Graphically, the initial plot (Figure 4a) suggests a weak relationship between "Production" and the fitted values from model (3). Consequently, the second boxplot (Figure 4b) reveals residual plots organized by country, which highlights the underlying issue of disregarding the impact of the country factor in data modeling.

Figure 4b reveals the most significant feature, indicating that residuals associated with the same country tend to have the same sign. Additionally, from a graphical standpoint, the time series variable lacks stationarity, implying that its mean and variance are not constant over time. Each year exhibits a distinct behavior over time, highlighting a limitation of the fixed-effects model, which assumes independence among observations of rare earth reserves. Therefore, there is a need to incorporate a "country effect" as a random effect in the model (4). This motivates the use of linear mixed-effects models. To account for the impact of the country and the effect of years within each country, a two-level cluster mixed model is considered. The first level represents countries (level 1), and the second level represents rare earth reserves and production (level 2). Each level contributes a distinct dimension to the analysis. The final model used in that section, corresponding to (Equation (4)), represents the k-th response ("Reserve or production"), denoted as $y_{ijk} = \log(Y_{ijk})$, for the j-th year in the i-th country. Formally, this can be expressed as follows:

$$Y_{ijk} = \beta_0 + \beta_1 \sin(w \times X_{ij} + T) + b_{1i} + b_{1(i)_j} + \varepsilon_{ijk} \tag{4}$$

In this model, the fixed effects include the intercept $\beta_0$, which represents the "standard" or average reserve or production at the initial time (Zero Time), and the slope $\beta_1$, which signifies the average increase in production or reserve per unit of time. The random effects are denoted as $b_{1i}$, capturing the variations in production or reserve among different countries (i), while $b_{1(i)_j}$ represents the random effects of the year factor within each country (i), capturing the annual variations. The term $\varepsilon_{ijk}$ denotes the within-individual errors.

To further exemplify the capabilities of this hierarchical data structure, we examine Table 2.

**Table 2.** Structured data with two levels: countries (Level 1) and production/reserve (Level 2).

| Countries | Years | Production/Reserve of REEs (t) | Times |
|---|---|---|---|
| | 1994 | 20,700 | 1 |
| United States | | | |
| | .. | … … … | .. |
| | 2023 | 43,000 | 29 |
| | 1994 | 30,600 | 30 |
| China | | | |
| | .. | … … … | . |
| | 2023 | 210,000 | 59 |

**Table 2.** *Cont.*

| Countries | Years | Production/Reserve of REEs (t) | Times |
|---|---|---|---|
| Australia | 1994 | 3300 | 60 |
| .. | .. | . . . . . . . . .. | .. |
| Brazil | 1994 | 400 | 100 |
| | 1994 | 5 | 361 |
| Other countries | . | . . . . . . . . | .. |
| | 2023 | 80 | 390 |

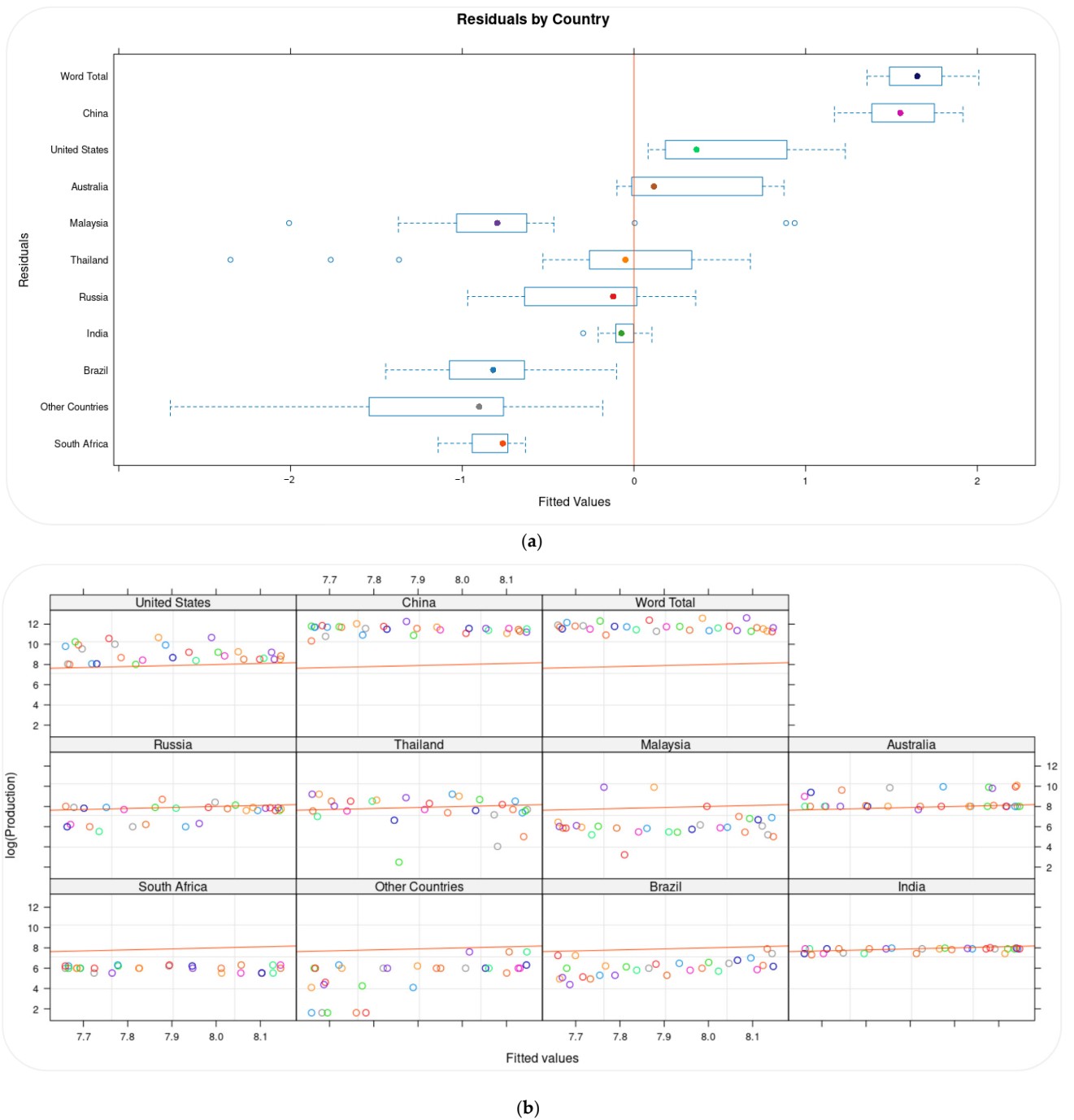

(**a**)

(**b**)

**Figure 4.** A scatterplot of log(Reserve) versus the fitted values, grouped by country (**a**), presenting boxplots of the residuals, also grouped by country (**b**) for model (3).

2.4.2. Model Selection Criteria

The Akaike Information Criterion (AIC) [94] is a method that assesses the probability of a model forecasting upcoming values based on in-sample fit. This technique, developed by Akaike in 1974, determines the most appropriate model by identifying the one with the lowest AIC value compared to other models [20]. The AIC can be employed to differentiate between the additive and multiplicative Holt–Winters models. On the other hand, the Bayesian Information Criterion (BIC), developed by Stone in 1979 [95], is another technique for model selection that balances the model's fit and complexity. A lower AIC or BIC value implies a better-fitting model [96,97]. The following (Equations (5) and (6)) are used to estimate the AIC and BIC [95] of a model:

$$AIC = -2LnL + 2K \tag{5}$$

$$BIC = -2LnL + 2LnN \times K \tag{6}$$

The value of the likelihood (L) in the formulation represents the likelihood of the observed data given the model parameters. The number of recorded measurements is denoted by N, and k represents the number of estimated parameters in the model. To test the stationary nature of the provincial laboratory test data, the R statistical package is utilized, as mentioned by Mohammed [98]. The data is then modeled using Holt–Winters and ARIMA models. For the Holt–Winters models, the parameters $\alpha$, $\beta$, and $\gamma$ are calculated by minimizing the one-step-ahead error, specifically the mean squared error (MSE) value, for each model. To compare the performance of the Holt–Winters models in estimating the provincial test volume, a set of ARIMA models is employed. The selection of the optimal model is based on criteria such as minimum Akaike information criteria (AIC) and Bayesian information criteria (BIC) values. Additionally, the model with the highest R-squared ($R^2$) value and the lowest AIC and BIC values is considered the optimal model.

2.4.3. R Software Programming Code

Statistical computations were conducted utilizing the statistical software R version 3.6.2. The mixed linear models were adjusted employing the "nlme" library [87]. The results include the computer programming code that describes the various R procedures employed for the analysis (Appendix A), as well as the modifications made to the original files.

**3. Results**

*3.1. Observation Data*

The analysis of REE reserves and mining data, obtained from the USGS, provides insights into the current status of global rare earth resources in major economies. Figures 5 and 6 visually illustrate the trends in rare earth reserves and mining activities over the specified time period, offering a clear representation of how these factors have evolved over the years. During the period of 2008–2022, the REE reserve exhibited fluctuations in various countries, with China maintaining its position as the leading country as shown in Figure 5. In 2008, China's rare earth reserve stood at 27 million metric tons, which increased to 55 million tons in 2010 but decreased to 44 million tons in 2023, accounting for 36.7% of the world's total. Vietnam ranked second with 22 million tons, followed by Brazil and Russia with 21 million tons and India with 690,000 tons. The United States held the seventh position with a reserve of 2.3 million tons, which remained relatively stable from 2008 to 2013 at 13 million tons, but declined to 1.8 million tons in 2014.

Figure 6 shows that the distribution of global REE resources is extremely uneven and illustrates the fluctuating trend in China's rare earth mining volume from 2008 to 2022. After experiencing fluctuations until 2014, China's mining volume began to steadily increase, reaching 105,000 tons in 2015 and peaking at 210,000 tons in 2022, accounting for approximately 60% of the global total. In 2018, due to factors such as Sino-US trade frictions, certain rare earth mining projects in the United States became active, leading to continuous growth in rare earth mining since then. By 2022, the United States would

account for 15.36% of the global total with 43,000 tons, followed by Australia, India, and Brazil. Overall, the global mining volume of rare earths increased, resulting in a decrease in China's share of rare earth resources in the world market.

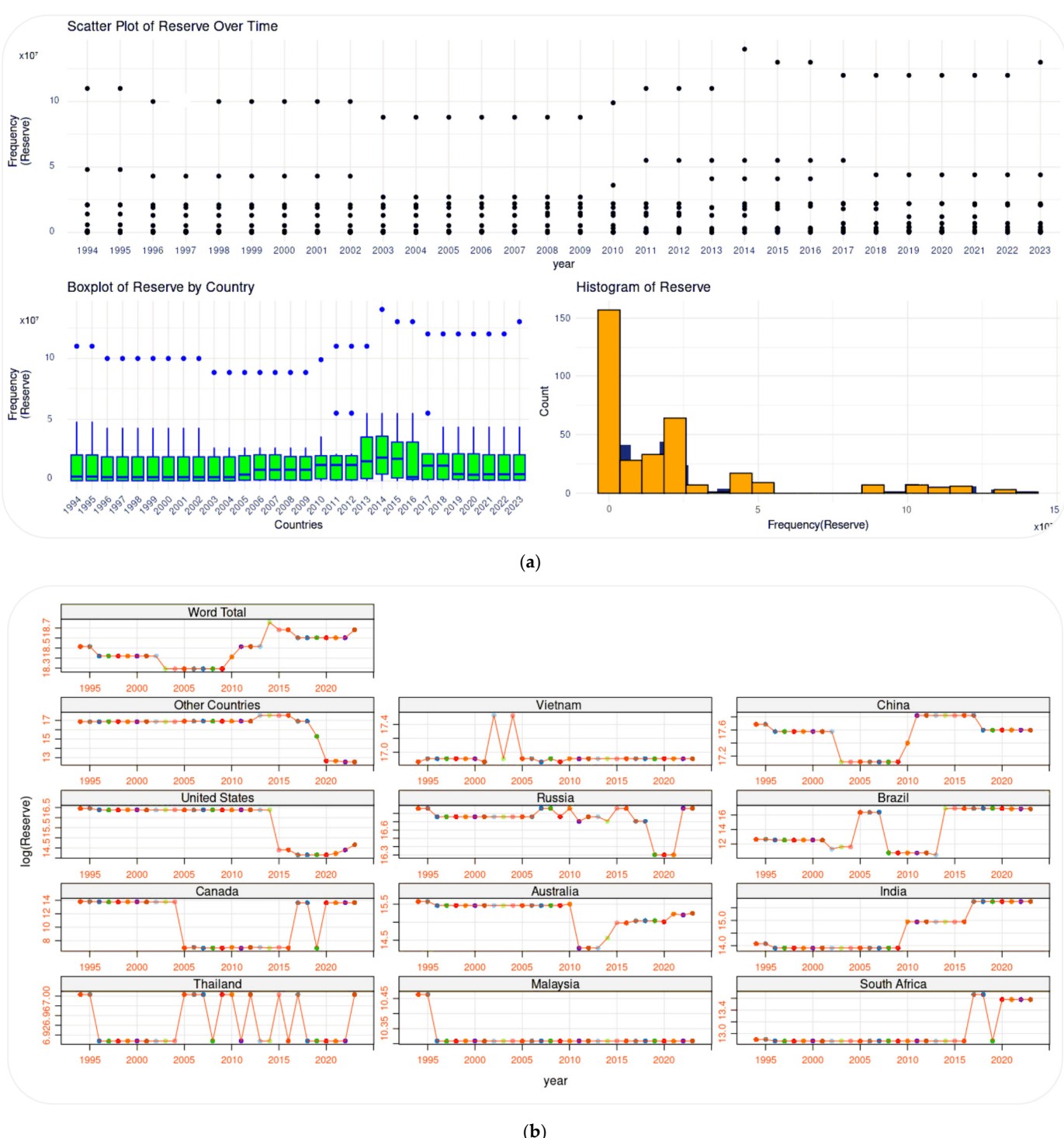

**Figure 5.** A depiction of the data reserve of rare earth by frequency (**a**) and country (**b**).

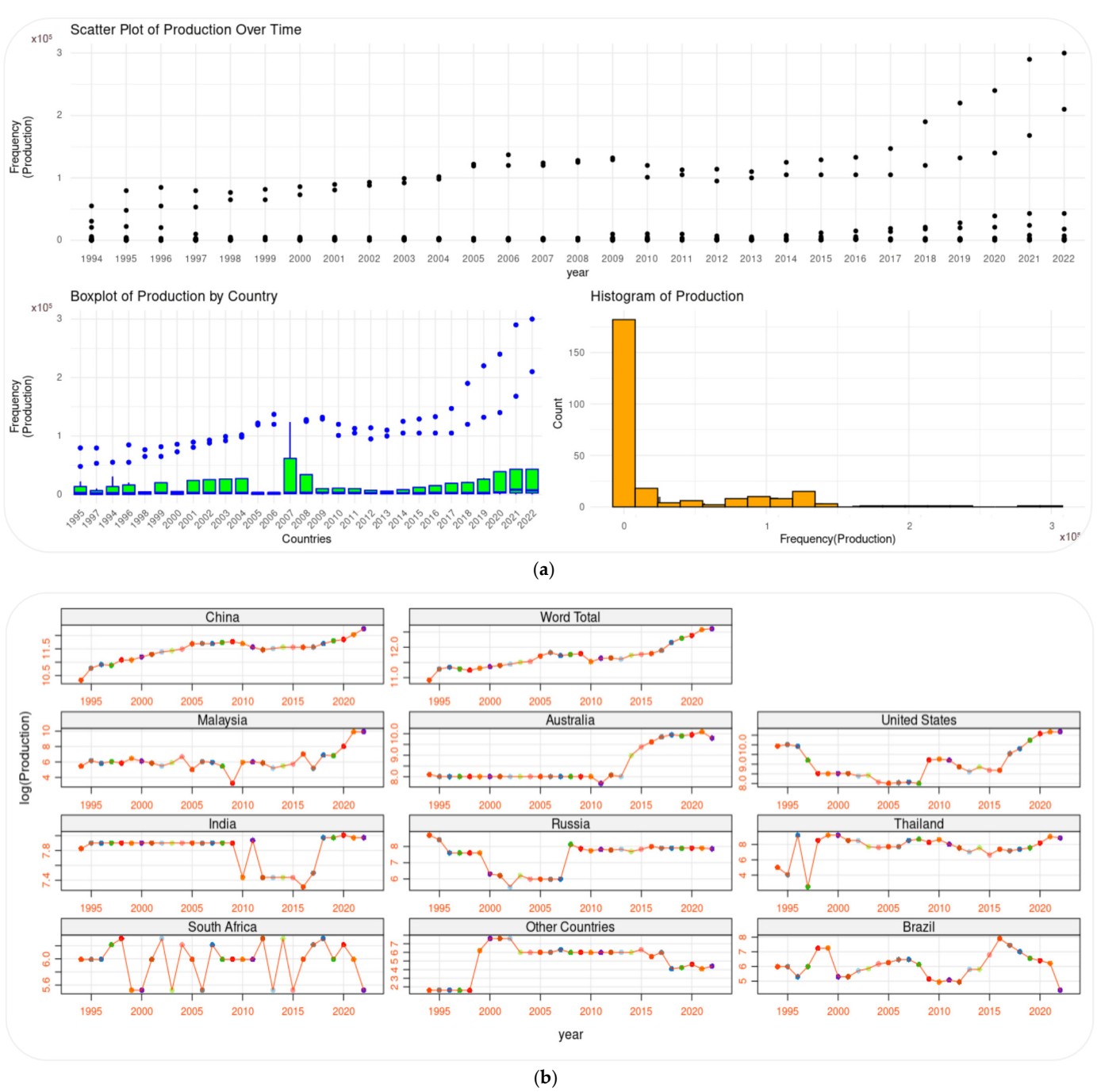

**Figure 6.** A depiction of the data production of rare earth by frequency (**a**) and country (**b**).

*3.2. Determining Missing Data Using the MICE Algorithm*

According to the input data, we have 56 missing values for the production and 44 missing values for the reserve. Therefore, Figure 7a,b show the values obtained from the MICE algorithm for the production and reserve in the study. Due to the random component of this algorithm, the procedure was applied five times for each missing value of the countries. The results of the MICE are presented in the figure in red color (Figure 7).

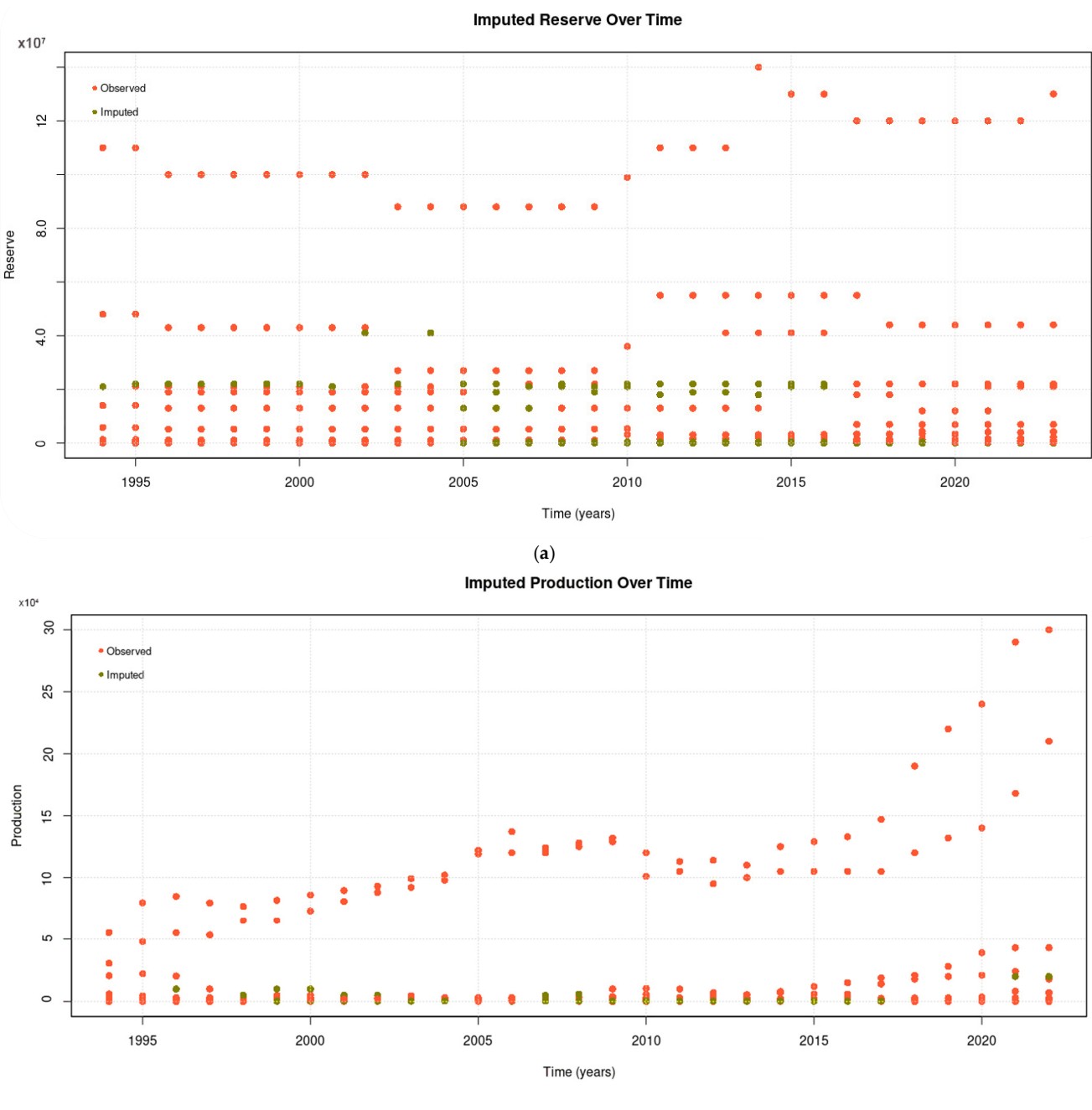

(**a**)

(**b**)

**Figure 7.** A depiction of missing values for reserve (**a**) and production (**b**) of rare earth using the MICE algorithm.

### 3.3. Adjustment of the Linear Regression Model (3)

Based on the input data, there are 56 missing values for production and 44 missing values for reserve. To address this, we applied the Multiple Imputation by Chained Equations (MICE) algorithm to impute these missing values. The MICE algorithm was run five times for each missing value of the countries to account for its random component. The imputed values obtained from the MICE algorithm for production and reserve are depicted in Figure 7a and Figure 7b respectively, highlighted in red color for clarity. The first model (Model 3) was adjusted, and its outcomes are presented in Table 2. The crucial factor to examine is the "*p*-value" for model validation, which should be below 0.05 (*p*-value < 0.05). It is evident from Table 1 that the parameter ($\beta_1$) fails to meet this criterion. Furthermore, in Figure 5, the inadequate alignment of values indicates that the applied model may not be

suitable for these particular samples. This misalignment can potentially lead to distortions in the obtained results.

### 3.4. Application of Model 3 in the Production of Rare Earth

Based on the findings from the adjustment of model (3) for production, as presented in Table 3, it is observed that the *p*-value for the parameter ($\beta_1 = 0.9469$) exceeds 0.05, indicating that it does not meet the required criterion. As a result, there is inconsistency in the average of the measurement errors. Furthermore, the lack of alignment in the values presented in Figure 8 indicates that the model is not appropriate for these specific samples. This lack of alignment is expected to lead to distorted results.

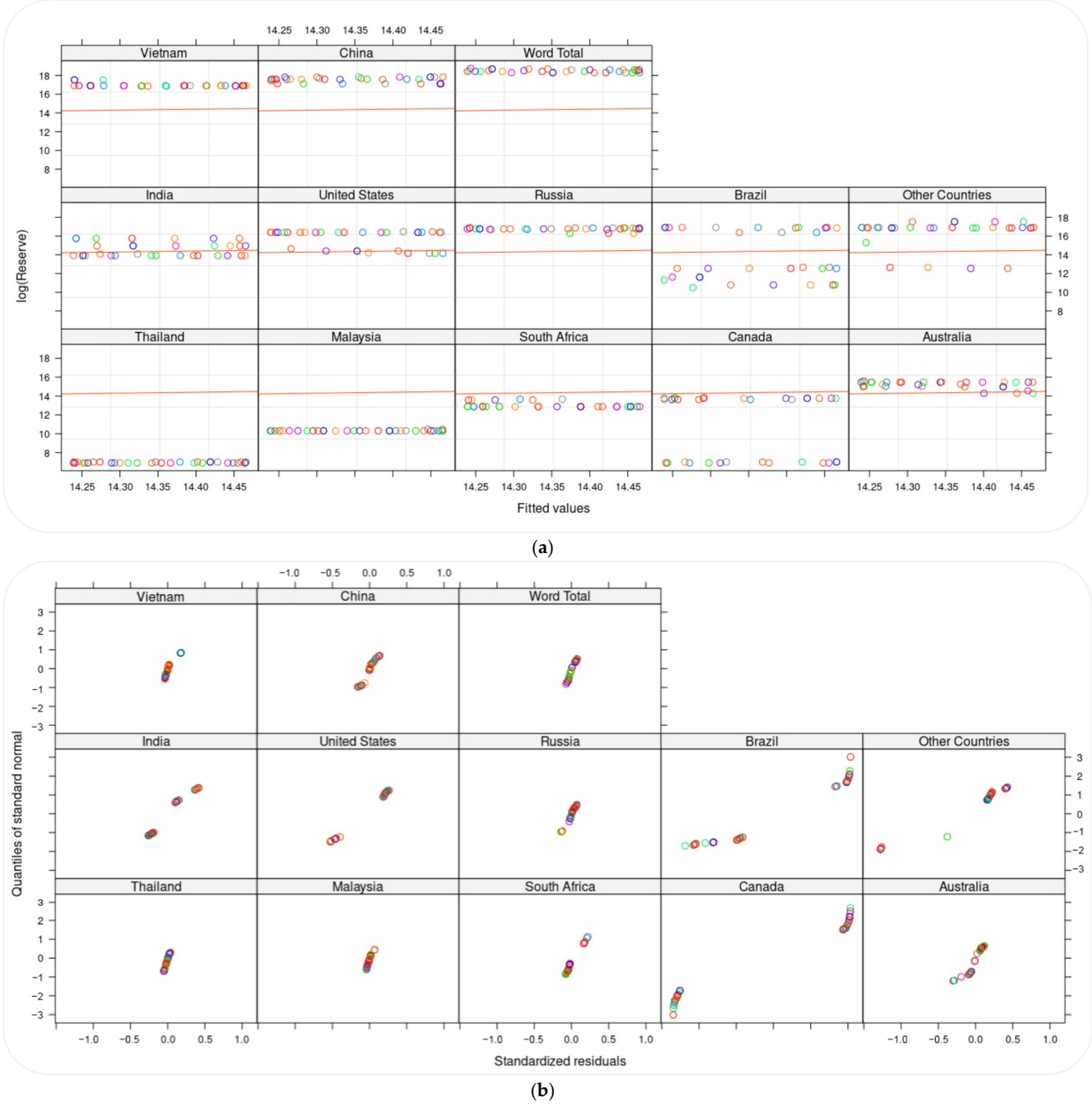

**Figure 8.** Adjustment of the model reserve, identity, normality, and effect of countries on residual values. (**a**) In mixed models incorporating random effects, the plot of responses versus fitted values serves as a crucial di-agnostic tool for evaluating model performance. This graphical representation

allows us to assess whether the model adequately captures the association between the independent variables and the dependent variable for each unit (country), thereby visualizing the accuracy of the model's predictions and uncovering any potential specification errors. An ideally fitted model is reflected in a scatter plot where data points are evenly dispersed around the 45-degree line, signifying a strong alignment between fitted and observed values. It's evident from our analysis that this criterion is met in our model. (**b**) In random-effects models, the "Standardized residuals" plot is employed to assess the normality of residuals, deter-mining if they conform to a normal distribution. This plot compares observed quantiles of residuals to the theoretical quantiles of a normal distribution. If residuals adhere to a normal distribution, points on the Standardized residuals plot should align approximately along a straight line, indicating a normal distribution with a mean of zero and constant variance. The alignment of points along a straight line suggests that residuals exhibit a normal distribution, ensuring the reliability of model results and adherence to underlying assumptions. In our case, this criterion is satisfied, affirming the normality of residuals and validating the robustness of our model outcomes.

**Table 3.** Summary table of the model (3) adjustment reserve case.

| Parameters | Estimation | | | | |
|:---:|:---:|:---:|:---:|:---:|:---:|
| | **Value** | **Std. Error** | **DF** | **t-Value** | ***p*-Value** |
| $\beta_0$ | 14.92 | 0.13 | 107.58 | 12.10 | 0.00 |
| $\beta_1$ | −0.01 | 0.19 | 307.00 | −0.06 | 0.94 |

*3.5. Estimation of Model 4 Reserve/Production*

Based on the results from the adjustment of model (4) presented in Tables 4 and 5, the *p*-value for both the fixed effects of production and reserve is less than 0.05. This indicates that the parameter values for the random effect and error are consistent. Consequently, to validate the model, we need to assess the fit of the adjusted values and errors. In terms of production and reserve, the data points are well-adjusted and contribute to normality, except for a few outliers in Thailand and other countries, as shown in Figures 6 and 7. Thus, the measurement errors also exhibit alignment in terms of their average. Consequently, the multilevel linear mixed-effects model is suitable for the specific sample we wish to analyze, as depicted in Figures 4 and 5.

**Table 4.** Summary table of the model (4) adjustment reserve case.

| Parameters | Estimation | | | | |
|:---:|:---:|:---:|:---:|:---:|:---:|
| | **Value** | **Std. Error** | **DF** | **t-Value** | ***p*-Value** |
| $\beta_0$ | 14.919150 | 0.6512213 | 376 | 909494 | 0.000 |
| $\beta_1$ | −0.106024 | 0.1133042 | 376 | −0.935748 | 0.035 |
| Random Effect | StdDev | | | | |
| $b_{1i}$ | 2.330446 | | | | |
| $b_{1(i)_j}$ | 1.432304 | | | | |
| $\sigma\Sigma$ | 0.6433784 | | | | |

**Table 5.** Summary table of the model (4) adjustment production case.

| Parameters | Estimation | | | | |
|:---:|:---:|:---:|:---:|:---:|:---:|
| | **Value** | **Std. Error** | **DF** | **t-Value** | ***p*-Value** |
| $\beta_0$ | 7.902929 | 0.6527760 | 307 | 12.106647 | 0.000 |
| $\beta_1$ | −0.192340 | 0.0786343 | 307 | −2.446013 | 0.015 |
| Random Effect | StdDev | | | | |
| $b_{1i}$ | 2.157274 | | | | |
| $b_{1(i)_j}$ | 0.8848628 | | | | |
| $\sigma\Sigma$ | 0.4324672 | | | | |

### 3.6. Adjustment of Model 4 Reserve/Production

After analyzing the results from the adjustment of model 4, as outlined in Tables 4 and 5, it is evident that the p-values for both the fixed effects of production and reserve are less than 0.05. This suggests that the parameter values for the random effect and error remain consistent. Consequently, in order to validate the model, it is imperative to evaluate the fit of the adjusted values and errors. With regards to production and reserve, the data points exhibit good adjustment and contribute to normality, with the exception of a few outliers observed in Thailand and other countries, as evidenced in Figures 9 and 10. As a result, the measurement errors also demonstrate alignment in terms of their average. Therefore, the multilevel linear mixed effects model appears to be appropriate for the specific sample we intend to analyze, as depicted in Figures 9 and 10.

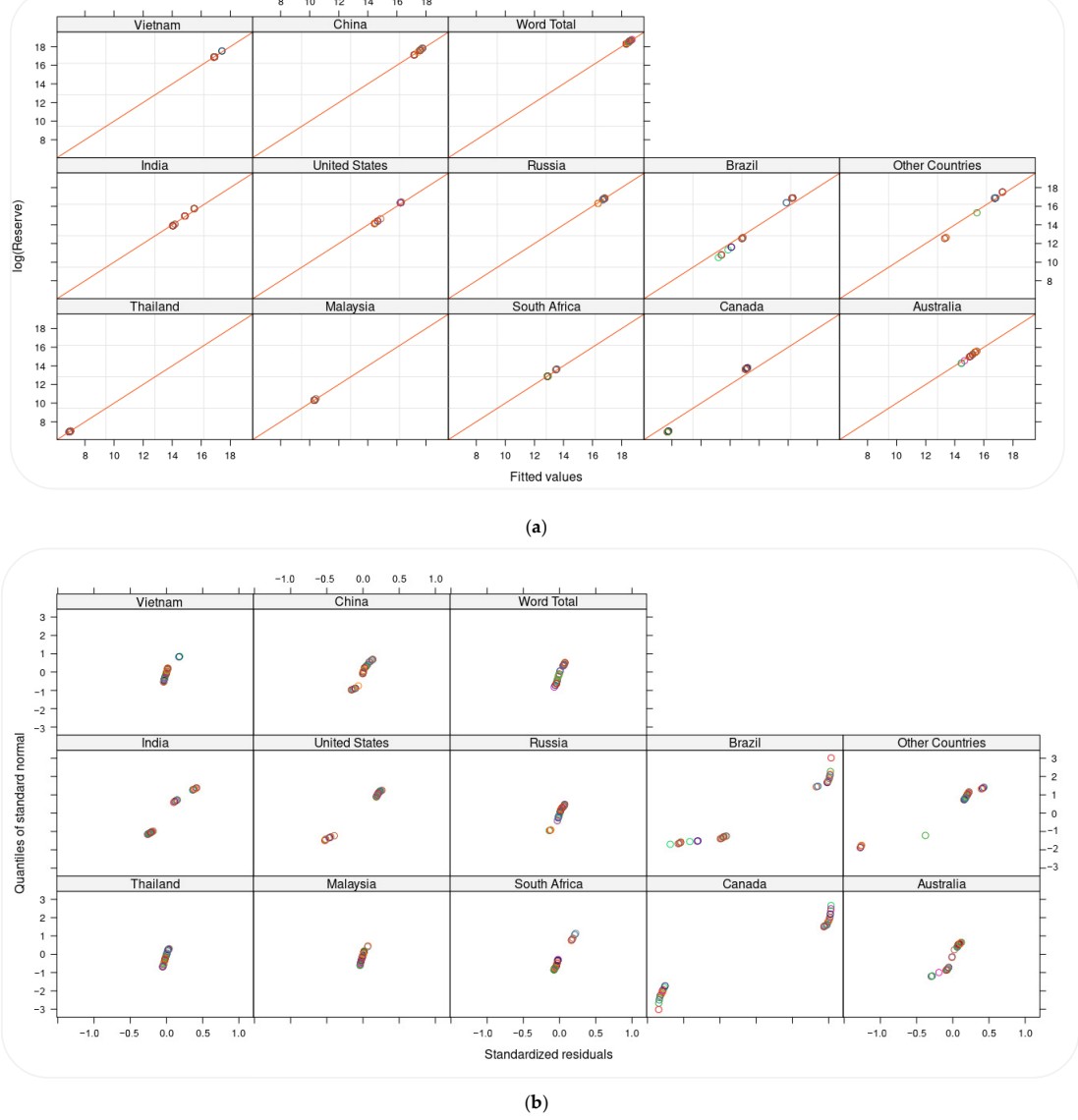

(**a**)

(**b**)

**Figure 9.** Adjustment of the model (LME) reserve, identity, normality, and effect of countries on residual values. (**a**) In mixed models incorporating random effects, the plot of responses versus fitted values serves as a crucial diagnostic tool for evaluating model performance. This graphical representation allows us to assess whether the model adequately captures the association between the independent variables and the dependent variable for each unit (country), thereby visualizing the accuracy of the model's predictions and uncovering any potential specification errors. An ideally fitted model is reflected in a scatter plot where data points are evenly dispersed around the 45-degree

line, signifying a strong alignment between fitted and observed values. It's evident from our analysis that this criterion is met in our model. (**b**) In random-effects models, the "Standardized residuals" plot is employed to assess the normality of residuals, determining if they conform to a normal distribution. This plot compares observed quantiles of residuals to the theoretical quantiles of a normal distribution. If residuals adhere to a normal distribution, points on the Standardized residuals plot should align approximately along a straight line, indicating a normal distribution with a mean of zero and constant variance. The alignment of points along a straight line suggests that residuals exhibit a normal distribution, ensuring the reliability of model results and adherence to underlying assumptions. In our case, this criterion is satisfied, affirming the normality of residuals and validating the robustness of our model outcomes.

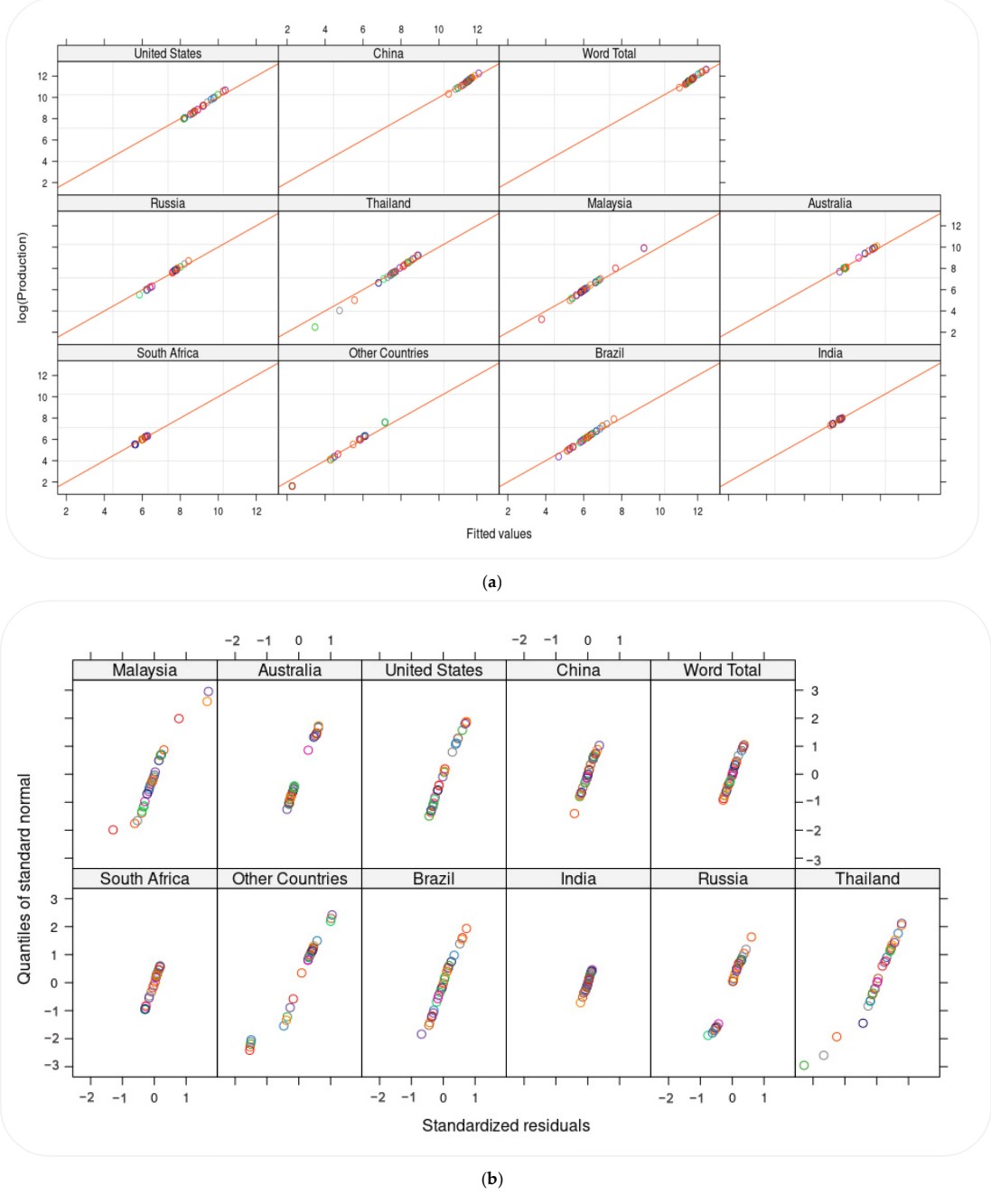

(**a**)

(**b**)

**Figure 10.** Adjustment of the model (LME) production, identity, normality, and effect of countries on residual values. (**a**) In mixed models incorporating random effects, the plot of responses versus fitted

values serves as a crucial diagnostic tool for evaluating model performance. This graphical representation allows us to assess whether the model adequately captures the association between the independent variables and the dependent variable for each unit (country), thereby visualizing the accuracy of the model's predictions and uncovering any potential specification errors. An ideally fitted model is reflected in a scatter plot where data points are evenly dispersed around the 45-degree line, signifying a strong alignment between fitted and observed values. It's evident from our analysis that this criterion is met in our model. (**b**) In random-effects models, the "Standardized residuals" plot is employed to assess the normality of residuals, determining if they conform to a normal distribution. This plot compares observed quantiles of residuals to the theoretical quantiles of a normal distribution. If residuals adhere to a normal distribution, points on the Standardized residuals plot should align approximately along a straight line, indicating a normal distribution with a mean of zero and constant variance. The alignment of points along a straight line suggests that residuals exhibit a normal distribution, ensuring the reliability of model results and adherence to underlying assumptions. In our case, this criterion is satisfied, affirming the normality of residuals and validating the robustness of our model outcomes.

### 3.7. Comparison between the Two Models (AIC/BIC)

When comparing the linear mixed-effects model (Model 2) and the linear regression model (Model 1), several statistical measures are used to determine the best model. These measures include the AIC (Akaike Information Criterion), BIC (Bayesian Information Criterion), and logLik (log-likelihood). The aim is to find a model with a low AIC and BIC, indicating a better fit, and a high logLik, suggesting a higher likelihood of the model accurately representing the data. The results showed that Model 2 exhibits lower AIC and BIC values than Model 1 (Table 6). Additionally, the logLik value for Model 2 is higher than that of Model 1. These findings indicate that Model 2 outperforms Model 1 regarding statistical goodness-of-fit measures.

**Table 6.** Comparison between the linear regression model and the multilevel linear mixed-effects model.

|  | Model | df | AIC | BIC | LogLik | Test | L. Ratio | *p*-Value |
|---|---|---|---|---|---|---|---|---|
| **Rep.gls** | 2 | 3 | 1440.6508 | 1451.9275 | −717.3254 | 1 vs. 2 | 484.4494 | <0.0001 |
| **Rep.lme** | 1 | 5 | 960.2014 | 978.9959 | −475.1007 | | | |

*Rep.gls: Repeated measures analysis using the generalized least squares (GLS) method. Rep.lme: Repeated measures analysis using linear mixed effects (LME) models.*

### 3.8. Production to 2051 and Reserve to 2053

The projections conducted by LME models for various economic countries have provided noteworthy insights into production and reserve trends up to 2051 and 2053, respectively. In the initial step, a selection decision was made between the classical simple regression model and the multilevel mixed-effects model, based on multiple comparison criteria, including AIC and BIC. Upon analyzing these criteria, it was determined that the mixed model is more effective than the conventional models because it was observed that the values for mixed models were lower compared to conventional models. Therefore, the mixed model was deemed more effective. In terms of model fitting, the classical model showed a *p*-value greater than 0.05 for the parameter $\beta_1$, which could potentially introduce bias to the results. Conversely, the mixed model demonstrated *p*-values less than 0.05 for both parameters $\beta_0$ and $\beta_1$. Furthermore, the mixed model successfully incorporated random parameters that were absent in the initial model, enhancing its accuracy.

The study findings indicate that the '$b_{1i}$' and '$b_{1(i)_j}$' values have a significant impact on the timing of peak production and reserve for REE. To ensure accurate theoretical predictions, it is crucial to determine suitable values for '$b_{1i}$' and '$b_{1(i)_j}$'. Higher '$b_{1(i)_j}$' values lead to a longer product life cycle for REE and result in a higher peak value. Conversely, techni-

cal factors, advancements in the REE end-use industry, and political factors will influence the intrinsic growth rate ('$b_{1i}$'), leading to an increase in peak production. These factors also affect the duration of resource exploitation, aligning with the scenario simulations conducted in this study. However, due to the uncertainties associated with these factors, future production and the timing of the peak may deviate from the presented predictions.

The following Figures 11a,b and 12a,b present the projected estimates for diverse annual production and reserves of rare earth elements conducted using the multilevel model.

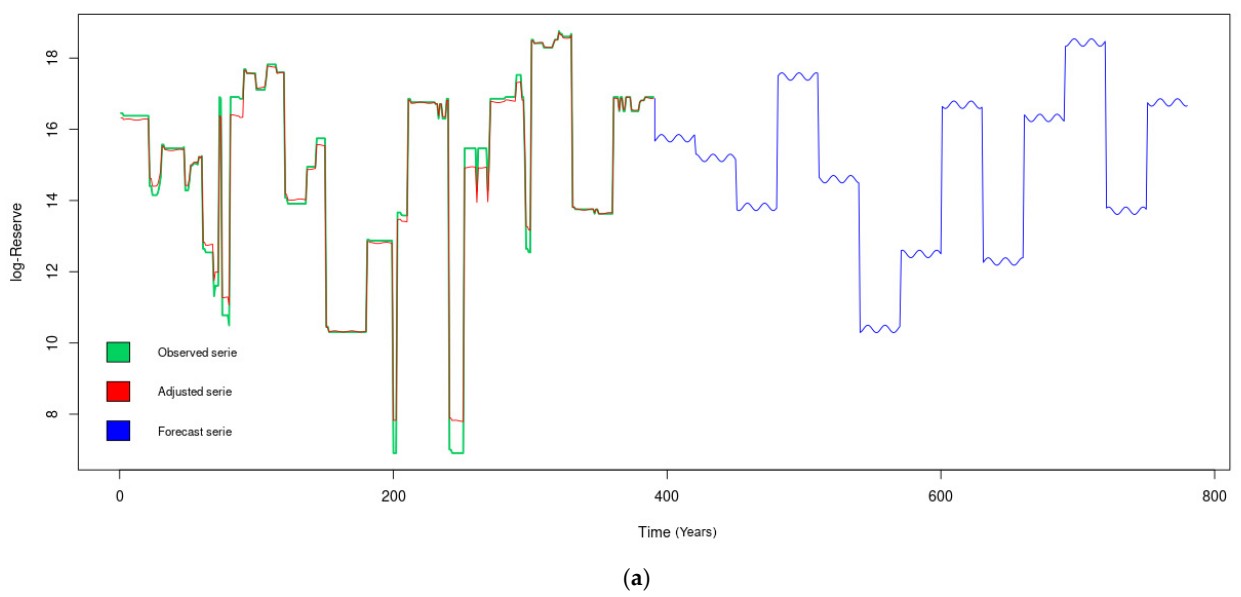

(**a**)

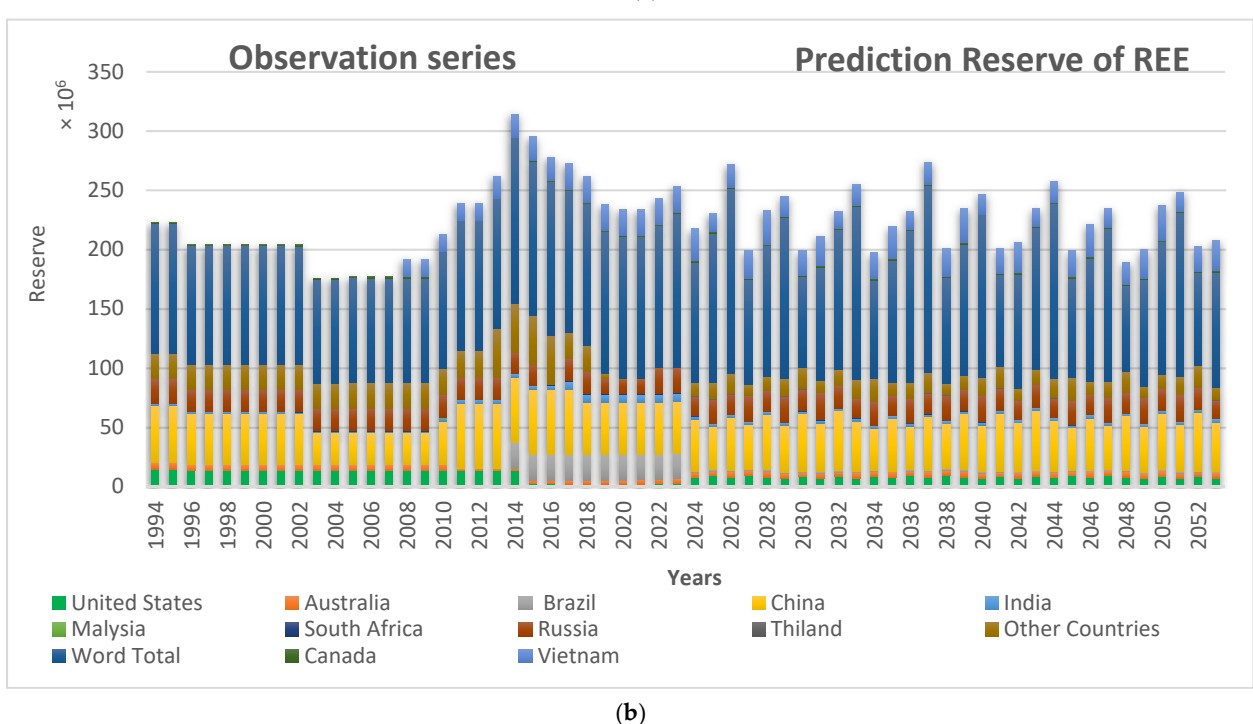

(**b**)

**Figure 11.** The annual reserve curve (2024–2053) (**a**) reserve annual different countries curve (2024–2053) (**b**).

The linear mixed-effects model's scenario simulations suggest that global production is set to increase, particularly during the period from 2024 to 2050. It is projected that peak production will occur in 2030, reaching 241,219.2665 tons. Another peak is expected in 2041, with a production mass of 247,752.76 tons. The final peak is anticipated in 2050, with a

production mass of 226,231.7743 tons. This increase is justified by the demand generated by the ambitious 2050 global wind-power targets, which cannot be met without expanding REE production by 11 to 26 times according to [99]. While China is expected to retain its position as the leading global producer of REE, the results showed that, by 2024, China will know a trend of growth. Peak production is projected to occur in 2030 at 190,606.4 tons, followed by another peak in 2041, with a production mass of 195,768.5 tons. According to the mixed-effects model scenarios, China's REE production will increase to 197,400 tons by 2022. Notably, the predicted annual data of rare earth production closely aligns with the actual data of 2022, with an average difference of less than 7%.

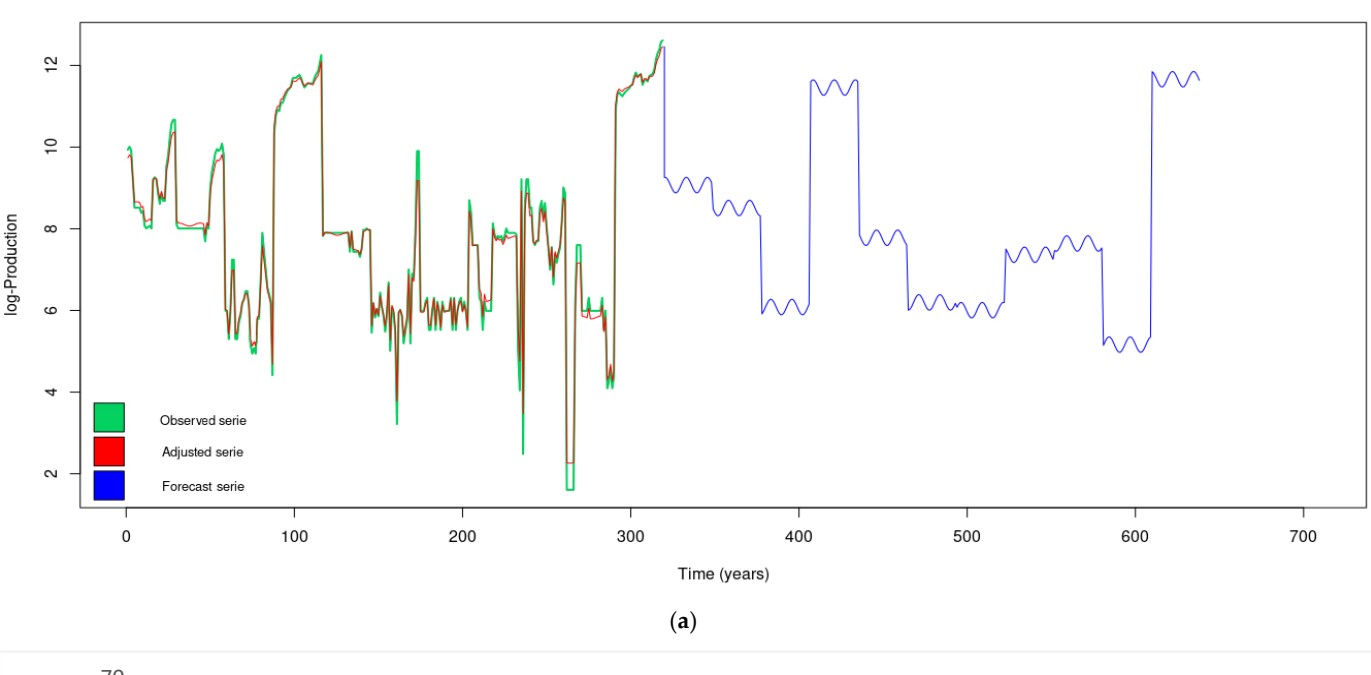

(**a**)

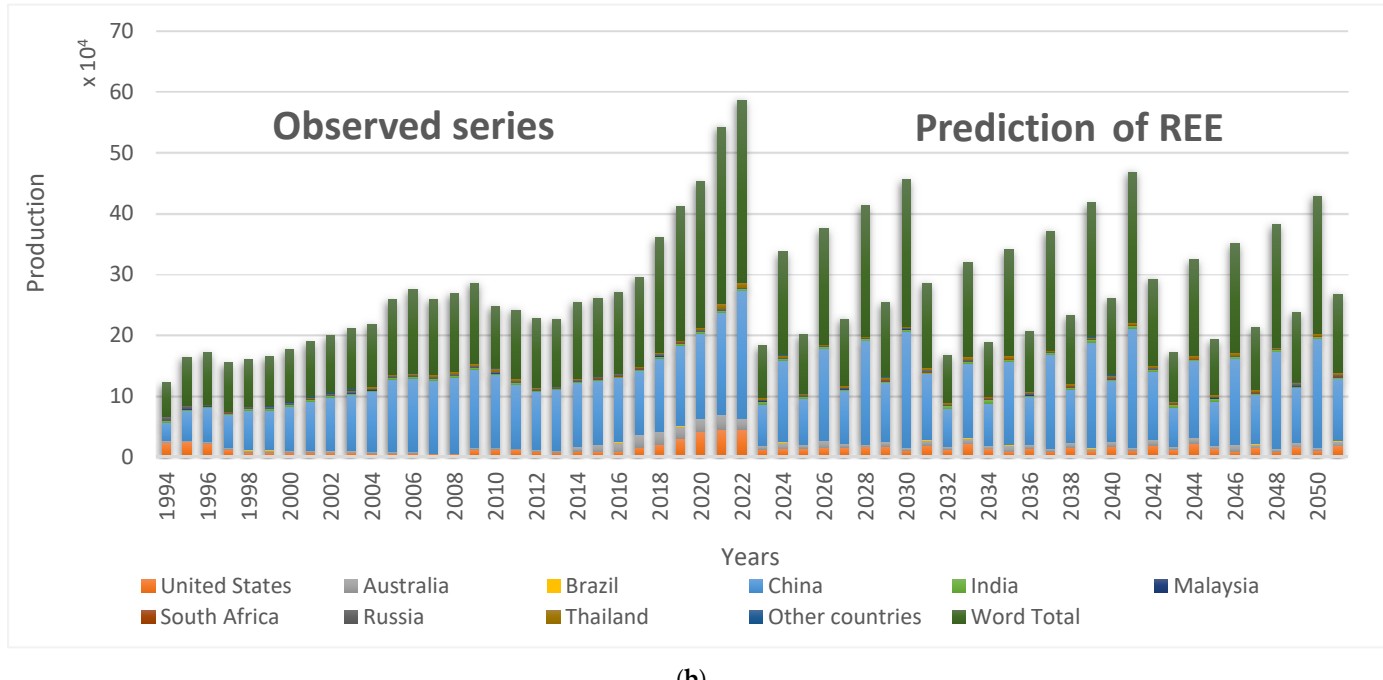

(**b**)

**Figure 12.** The production annual curve (2023–2051) (**a**) production annual of different countries curve (2023–2051) (**b**).

The United States, with a high production rate, is expected to become the second-largest producer of REEs. Its peak production is projected to occur in 2033, with a mass

of 19,009.6745 tons (Figure 12b). Another peak is anticipated in 2044, with a production mass of 19,524.51 tons. Australia, securing the third position, is projected to reach its peak production in 2033, with a mass of 11,534.8478 tons. However, the lowest production is expected to be observed in 2037, with a mass of 3428.188 tons. Following this, India, Thailand, and Russia are also expected to contribute to REE production. India is projected to reach its peak production in 2023, with a mass of 5128.5758 tons. Thailand's peak production is anticipated in 2024, with a mass of 4218.9158 tons, while Russia's peak production is expected in 2033, with a mass of 3619.9535 tons. Another peak in production is anticipated for these countries in 2045, 2046, and 2042, respectively. During this period, India is projected to produce 5410.1315 tons, Thailand 2257.3599 tons, and Russia 3305.5146 tons.

Furthermore, Malaysia, Brazil, and South Africa are expected to have a lower volume of production compared to the leading producers. The first peak in production is anticipated in these countries in 2027, with Malaysia projected to produce 2707.2517 tons, Brazil to produce 473.6998 tons, and South Africa to produce 750.8331 tons. A second peak in production is anticipated in 2038. Notably, the timing of the last peak varies among the three countries, occurring in the periods 2047, 2050, and 2051. During these peaks, Malaysia is projected to produce 1013.0432 tons, Brazil 985.2022 tons, and South Africa 890.8896 tons.

The findings indicate that global reserves are projected to decline, particularly from 2030 to 2048 overall. The peak reserve is expected to transpire in 2026, reaching a mass of 156,926,006.93 tons. Another peak is anticipated in 2037, with a reserve mass of 158,257,305.14 tons. Conversely, a lower reserve was observed in 2030, with a mass of 77,175,546.45 tons. Another lower peak was observed in 2048, with a reserve mass of 72,866,572.14 tons. The decrease in reserves observed between 2030 and 2042 can be linked to increased demand for products during those years, showcasing a direct correlation between reserve levels and the production of rare earth elements (REEs) globally. This correlation is rooted in the significant demand for REEs across various applications, driven primarily by ambitious global wind-power objectives and the growing adoption of electric vehicles. Consequently, there is a pressing need for an 11- to 26-times expansion in REE production to meet this heightened demand, consequently affecting reserve levels.

Despite the anticipation of high rare earth element production in China, the country is still expected to maintain its position as the primary global reserve for REE, representing 39% of the total reserve. The peak reserve is projected to be reached in 2032, amounting to 50,474,342.1 tons, followed by another peak in 2043 with a reserve mass of 50,902,546.47 tons, attributed to the discovery of new mines. Notably, the years with higher production demonstrate a lower reserve. Additionally, Vietnam secures the second spot in world reserves, with a projected peak of 50,474,342.1 tons in 2028, followed by another peak in 2039 with a reserve mass of 50,902,546.47 tons. The last peak is expected in 2050. It is worth mentioning that lower reserve masses are expected in 2025 and 2043, with a further decrease to 14,236,573.64 tons in 2032.

Russia holds the third position, projected to reach a peak reserve of 22,907,990.73 tons in 2031, followed by another peak in 2051 with a reserve mass of 22,448,129.24 tons. The lower reserve masses are expected in 2042 and 2053, with amounts of 15,610,028.88 tons and 15,742,458.19 tons, respectively. On the other hand, the United States and Australia will secure the fourth and fifth positions. In the US, the peak reserve is projected to be reached in 2027 and 2042, with masses of 8,983,237.05 tons and 9,059,447.29 tons, respectively. As for Australia, the maximum reserve is expected in 2039 with a mass of 5,170,508.5 tons, followed by the second peak in 2050 with a reserve of 5,214,373.05 tons. Lower reserve volumes were observed in 2030 and 2052 in both countries.

In comparison to the leading reserve countries for rare earth elements, India, Brazil, Canada, South Africa, Thailand, and Malaysia are projected to have lower volumes of rare earth reserves, respectively. Among these countries, India, Canada, and Malaysia are expected to experience their first peak in production in 2025, with projected reserves of 2,831,677.08 tons, 1,372,631.43 tons, and 40,973.38 tons, respectively. A second peak in production is anticipated in 2036, with volumes of 2,855,699.91 tons, 1,384,276.29 tons, and

41,320.98 tons. These countries are also expected to have lower reserve mass in 2029 and 2048 (Figure 11a).

Brazil is projected to reach a peak reserve of 1,293,008.71 tons in 2031, followed by another peak in 2042 with a reserve mass of 1,303,978.08 tons. The lower reserve volumes are expected in 2033 and 2045, amounting to 906,763.11 tons. In South Africa and Thailand, higher reserves are anticipated in 2039 and 2051, with mass of 355,038.06 tons and 265,687.22 tons, respectively. The second projected peak for South Africa is 2,051,299,840.26 tons while, for Thailand, it is 344,653.24 tons. Conversely, the lower reserve in South Africa is expected in 2041, totaling 248,981.63 tons while, in Thailand, the lower peak will be observed in 2046 with a mass of 168,073.14 tons.

The increase in rare earth element (REE) reserves is attributed to the vital role these elements play in the development of nations. Following the 2010 China crisis, which highlighted the vulnerability of relying on a single source for Chinese REEs, countries have been compelled to explore and discover new sources and mines for these crucial minerals. Consequently, the model predicts a surge in reserves in major countries, characterized by three distinct phases of growth.

Based on the literature review, it is identified that there is only one study that analyzes the long-term production of rare earths worldwide using quantitative methods [17]. In addition, there are three studies that present the long-term production trajectory of China's rare earths [41,49,77].

Furthermore, the Mineral Commodity Summaries 2018 report published by the USGS reveals that, in 2017, the total global rare earth production amounted to 133.5 thousand tons, with 78.65% of this production originating from China. This paper assumes that this proportion remains constant in the future. Therefore, the world's rare earth production can be estimated based on the results of these three studies by dividing their production figures by 78.65%.

Our study corroborates previous research on global rare earth production using the curve-fitting model and the Richards method, as presented by Wang [17]. It suggests an upward trend in global rare earth production, particularly from 2024 to 2050. However, there are contrasting views presented by Wang [41,77], who proposes a different pattern. According to their studies, rare earth production is projected to initially experience rapid growth, reach a peak, and subsequently decline significantly by 2050, supporting the hypothesis of lower production in the long term. Regarding China's rare earth production, our study aligns with previous research that utilized the Wang model [77]. Our results indicate a growth trend, with projected peak production in 2030 at 190,606.4 tons, followed by another peak in 2041 with a production mass of 195,768.5 tons. These findings are consistent with Wang's study, which also suggests a steady annual growth rate of 1.72% from 2024 to 2045. However, it is important to note that the study forecasts a slight decrease in production, with an estimated annual decline of 4% by 2050. This decrease represents the smallest reduction in production compared to earlier years.

The history and present state of the REE supply chain exhibit the important role these materials already play in the world economy. Projections of a sharp increase in demand over the coming decades raise several questions about the future supply risks to this industry. A 2012 MIT study by Alonso et al. thoroughly explores this question of future supply, and projects total global demand up to 2035 under five divergent scenarios [100].

One of these scenarios uses the IEA Blue Map scenario to estimate future automotive electrification (Hub, n.d.). This model only seeks to reduce global carbon emissions by 50% by 2050.

Under this scenario, the study projects that, by 2035, global demand for REEs will reach close to 450,000 tons per year, compared to approximately 200,000 tons per year in 2021 (USGS 2021). This represents more than a doubling in the size of the industry in just 15 years. Furthermore, the rate of demand growth accelerates rapidly, as do projections of EV production up to 2050, indicating that this increase in industry demand is only the beginning of a pattern of accelerating growth that will likely last for decades [54].

The demand for REEs in EVs under the pessimistic, neutral, and optimistic scenarios will be 44 thousand tons, 89 thousand tons, and 179 thousand tons, respectively. The demands for Pr, Nd, and Dy will be higher than the demands for the other elements. The demands for these three elements in EVs will be 6–23 thousand tons for Pr, 26–108 thousand tons for Nd, and 11–47 thousand tons for Dy in China up to 2030, see for example Leal [24]. This is why it provides an encompassing understanding of the potential constraints resulting from rare earth element depletion/production and their impact on the energy sector.

## 4. Limitations and Future Study

In practical terms, the production and reserves of REEs can be influenced by a multitude of factors. While this paper provides an encompassing understanding of the potential constraints resulting from rare earth element depletion, the forecast results presented here predominantly rely on a geological perspective supported by data from the USGS. However, due to data limitations, there is a dearth of quantitative analysis concerning the impact of environmental factors on future production projections. Moreover, the current global significance of rare earth elements and their uneven distribution underscore the potential influence of political issues on production, which remains unaddressed in this paper. Additionally, conducting quantitative assessments to evaluate the effects of mining investments and recycling on the supply of rare earth elements would be highly advantageous. It is imperative that future studies comprehensively address these matters.

## 5. Conclusions

The higher consumption of rare earths is currently considered a major concern for the green industry. Therefore, MLE models are statistical approaches that enable the analysis of data with multiple levels of variation and take into consideration both fixed effects, which influence the average level of the model, and random effects, such as rare earth reserve data from different countries and globally, which impact the model's variance. These models have the potential to identify trends and patterns in rare earth reserves and production, allowing for predictions of future reserves based on factors like mining technology, environmental regulations, and demand for rare earth elements. Simultaneously estimating the fixed and random components of the model leads to more precise modeling and satisfactory results. The model consistently produces outcomes for production and reserves, with minimal standard deviation values in both fixed and random components. Additionally, the "*p*-values" are strictly below 0.05, indicating statistical significance. The production, reserve, and errors align well with the adjusted values, confirming the model's suitability. Future production and reserve projections carried out by LME models for the economic countries have shown significant results up to 2051 for production as well as for the reserve up to 2053. They reveal declining reserve levels in certain countries and significant increases in others, accompanied by a substantial rise in production during these years. These findings suggest the possibility of depletion periods in certain countries. We can draw two major conclusions from these analyses.

1.  In 2041, global REE production is anticipated to reach a higher level, reaching 247,752.76 tons. China is expected to maintain its position as the leading producer, accounting for 79% of global production with a mass of 195,768.5 tons. The United States is projected to become the second-largest producer, reaching a production mass of 19,524.51 tons by 2044. Australia is expected to secure the third position, with its peak production estimated to occur in 2033 at 11,534.8478 tons. Additionally, India, Thailand, Russia, Malaysia, Brazil, and South Africa are anticipated to contribute to the global rare earth production, albeit at lower volumes compared to the leading economies.
2.  The model indicates a projected decline in the global reserve of rare earth elements, particularly from 2030 to 2048. The peak reserve is expected to occur in 2037, reaching a mass of 158,257,305.14 tons. China holds a clear monopoly as the primary global

reserve for REEs, accounting for 39% of the total reserve. Its reserve mass is projected to be 50,902,546.47 tons in 2043. Vietnam secures the second position in world reserves, with a projected reserve mass of 50,902,546.47 tons in 2039. Russia holds the third position and is expected to reach a peak reserve of 22,907,990.73 tons by 2031. The United States and Australia secure the fourth and fifth positions, respectively. Following them are India, Canada, Malaysia, Brazil, South Africa, and Thailand in terms of reserve volumes.

This study holds great value for policymakers, industry leaders, and investors who face decisions concerning the future of rare earth elements. It can provide insights into identifying countries or regions at risk of significant reserve depletion, aiding in informed investment choices for new mining projects. Moreover, it plays a crucial role in assessing the potential impact of increased production on global economics and human well-being. Nevertheless, it is essential to acknowledge the inherent uncertainty in predicting the future of rare earth elements due to numerous unpredictable factors. Furthermore, any comprehensive study of this nature must encompass the environmental and social implications of rare earth mining and processing, emphasizing the need for sustainable and responsible practices within the industry.

**Author Contributions:** Conceptualization, H.E.A., E.K.C. and E.M.A.; methodology, E.K.C., E.M.A. and H.E.A.; software, E.K.C. and H.E.A.; validation, H.E.A., E.M.A., Y.O.L., E.M.A., E.K.C. and F.C.; formal analysis, E.K.C., H.E.A., Y.O.L., R.E.H., E.M.A. and F.S.; writing—original draft preparation, E.K.C., E.M.A., F.C. and H.E.A.; writing—review and editing, E.K.C., H.E.A., Y.O.L., E.M.A., R.E.H., F.C. and H.E.A.; visualization, H.E.A., E.K.C. and E.M.A.; supervision, E.K.C., F.C. and F.S.; funding acquisition, F.C. All authors have read and agreed to the published version of the manuscript.

**Funding:** This research received no external funding.

**Institutional Review Board Statement:** Not applicable.

**Informed Consent Statement:** Informed consent was obtained from all research subjects.

**Data Availability Statement:** The authors strongly encourage interested researchers to contact us, as we are more than willing to share the data upon request.

**Acknowledgments:** The authors would like to thank all those who collaborated in this work on the field sampling, laboratory analysis, and writing manuscript teams from the Laboratory of Physical Chemistry of Materials, Natural Substances and Environment, MARETEC, and the Department of Mathematics and Statistics of FST, Tangier, in Morocco.

**Conflicts of Interest:** The authors declare no conflicts of interest.

## Appendix A

```
#
#---------------- installing and loading packages
#
if(!require("RcmdrMisc")){install.packages("RcmdrMisc")};library(RcmdrMisc)
if(!require("tidyr")){install.packages("tidyr")};library(tidyr)
if(!require("mice")){install.packages("mice")};library(mice)
library(nlme)
library(lattice)
#if(!require("flextable")){install.packages("flextable")};library(flextable)
if(!require("tseries")){install.packages("tseries")};library(tseries)
if(!require("caschrono")){install.packages("caschrono")};library(caschrono)
if(!require("forecast")){install.packages("forecast")};library(forecast)
if(!require("tactile")){install.packages("tactile")};library(tactile)
if(!require("ggplot2")){install.packages("ggplot2")};library(ggplot2)
if(!require("gridExtra")){install.packages("gridExtra")};library(gridExtra)
```

```
if(!require("lqmm")){install.packages("lqmm")};library(lqmm)
#if(!require("qrLMM")){install.packages("qrLMM")};library(qrLMM)

#
#--------------- Importing data
#
donAmBer <-readXL("F:/2023/DocAmiBeroho/donAmBer1.xlsx", rownames=FALSE,
header=TRUE, na="", sheet="PRODUCTION RRE", stringsAsFactors=TRUE)
head(donAmBer)
names(donAmBer)=c("countrys",1994:2022)
head(donAmBer,2)
donAmBer1=donAmBer
head(donAmBer1)

#
#--------------- Long data transformations
#
donAmBer1.L=as.data-1.frame(donAmBer1%>% pivot_longer(cols=names(donAmBer1)[-1],
names_to='year',values_to="Reserve"))
summary(donAmBer1.L)
#
#--------------- data organization
#
#donAmBer1.L$year=as.numeric(donAmBer1.L$year)
donAmBer1.L$countrys=as.factor(donAmBer1.L$countrys)
summary(donAmBer1.L)
head(donAmBer1.L)

#---------------------
#--------------------- Data Exploration#---------------------

p1 <- ggplot(donAmBer1.L, aes(x =year, y =Reserve)) + geom_point() +geom_smooth(method =
"lm", se = F, color = "red", linetype = "dashed") +theme_bw() + labs(y = "Frequency\n(Reserve)")
p2 <- ggplot(donAmBer1.L, aes(x = reorder(year, -Reserve), y =Reserve)) + geom_boxplot()+
theme_bw() + theme(axis.text.x = element_text(angle=90)) +labs(x = "countrys", y =
"Frequency\n(Reserve)")
p3 <- ggplot(donAmBer1.L, aes(Reserve)) +geom_histogram() + theme_bw() + labs(y = "Count", x
= "Frequency(Reserve)")
grid.arrange(grobs = list(p1, p2, p3), widths = c(1, 1), layout_matrix = rbind(c(1, 1), c(2, 3)))

#--------------- Check for missing values
#
sum(is.na(donAmBer1.L))
sapply(donAmBer1.L, function(x) sum(is.na(x)))

#
#------------- Now that the dataset is ready for imputation,
#------------- we will call the mice package.
#
donAmBer.L.mice = mice(donAmBer1.L, m = 5, maxit = 50, meth = "pmm", seed = 123)
#
#------- Create a dataset after imputation.
#
donAmBer.L.impu <- complete(donAmBer.L.mice)
```

```
#
#--------------- Check for missing values
#
sapply(donAmBer.L.impu, function(x) sum(is.na(x)))
summary(donAmBer.L.impu)
#
# ----- Data graph with missing values (1)
#
plot(donAmBer.L.impu$year, donAmBer.L.impu$Reserve, col = mdc(1:2)[1 +
is.na(donAmBer1.L$Reserve)], xlab = "Time(years)", ylab = "Reserve")

#
#------------------------Exploration by countrys: (2) -----
#
donAmBer.L.impu$year=as.numeric(donAmBer.L.impu$year)
xyplot(log(Reserve)~year|countrys, type=c("g","p", "l"), pch=16, auto.key=list(border=TRUE),
par.settings=simpleTheme(pch=16), scales=list(x=list(relation='free'), y=list(relation='free')),
data=donAmBer.L.impu,index.cond = function(x,y)max(y),layout=c(3,5))
#
#--------------------- Modelization
#
donAmBer.L.impu$Time=seq(1:length(donAmBer.L.impu$year))
donAmBer.L.impu1=groupedData(log(Reserve)~Time|countrys,donAmBer.L.impu)

Rep.gls=gls(log(Reserve)~sin((0.5)*Time+pi/2),data=donAmBer.L.impu1)
Rep.lme=lme(log(Reserve)~sin((0.5)*Time+pi/2),random=~1|countrys/year,data=
donAmBer.L.impu1)
anova(Rep.lme, Rep.gls)
summary(Rep.lme)
#
#--------Validation
#::::::::::::::.::: observed responses versus fitted values
#
plot(Rep.lme,log(Reserve)~fitted(.)|countrys,abline=c(0,1),layout=c(3,5),
col = "black")
#
# ::::::::::::: Normality by countrys
#
qqnorm(Rep.lme,~resid(.,type = "p")|countrys,col = "black")
#::::::: generate diagnostic plots: fitted values by countrys
plot(Rep.lme, countrys~resid(., type = "p"), abline = 0)
#------------------
#------- Forecast |----------------Forecast on the new sample
#------------------

#
#--------------------------------------------
#. . . . . . . . . . .. ... Graphs . . ..
#--------------------------------------------
#
plot(donAmBer.L.impu1$Time,log(donAmBer.L.impu1$Reserve),xlim=c(min(donAmBer.L.impu1
$Time),max(temps_pr)),ylab="log-Reserve",xlab="Time",type="l")
#
# ---------lines for modèle ajusté-----------
#

fit=fitted(Rep.lme)

lines(donAmBer.L.impu1$Time,fit,col="red")
```

```
#
#-------------- forecast time
#

temps_pr=seq(from=length(donAmBer.L.impu1$Time)+1,to=length(donAmBer.L.impu1$Time)+
319,by=1)
year.f=seq(max(donAmBer.L.impu1$year)+1,to=max(donAmBer.L.impu1$year)+length(unique
(donAmBer.L.impu1$year)),by=1)
n.rep=length(donAmBer.L.impu1$year)/length(unique(donAmBer.L.impu1$year))
#donAmBer.L.impu1$year=rep(year.f,n.rep)
new.donAmBer.L.impu1 <- data.frame(year=rep(year.f,n.rep),Time=temps_pr,countrys =
donAmBer.L.impu1$countrys)
pr=predict(Rep.lme, new.donAmBer.L.impu1, level = 0:1)
print(pr)
lines(c(temps_pr [1]-1,temps_pr),c(fit[length(fit)],pr$predict.countrys),col="blue")
legend(x=350,y=4,legend=c("Observed serie","Adjusted serie","Forecast serie"),fill=c("black",
"red", "blue"))
```

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
