# Peer review of "Predicting the Production and Depletion of Rare Earth Elements and Their Influence on Energy Sector Sustainability through the Utilization of Multilevel Linear Prediction Mixed-Effects Models with R Software"

_sustainability, doi:10.3390/su16051951_

Round 1

Reviewer 1 Report

Comments and Suggestions for Authors

Detailed comments are on the marked PDF.

Main revisions need are clearer figures that are publication quality.

In places the MS does well to be about te sustainability of REEs, but in other places its a modelling MS instead and not clear - the worst example is Figure 11, where the x-axis makes no sense - what are the units, they go to 700 for a ~25 year time span. The y-axis is not scientific units, and the real world meaning and likelohood of reserves rising needs to be discussed in the text.

Another example about changing the paper from modelling paper to sustainbility paper is examining different cases - e.g. I think Figure 111 should have versions where PRC has ~79% of the REE as reserves, as well calculations for 70 and 85%.

The MS also mentioned codes and scripts, but I could not see these with the document and could not test it, neither therefore could a reader.

This needs to be corrected before publication

Finally the figures need finishing to publication quality. Editting output from R can be a pain. 2 options are:

1) Edit PDFs in adobe editor or AI

2) SVGs and edit these using a free svg editor like inkscape.

This is the easy way to correct things like variable names on graphs that have odd capitalisation or special characters etc. The other options are lots of scripting in GGplot

It doesn't matter which way its done, but graphs shoudl  have all elements visible at 100% screen size.

This is a good study, but it needs heavy revision before it can be published.

Reviewer 2 Report

Comments and Suggestions for Authors

The submitted manuscript forecasts the production and depletion of rare earth elements. The manuscript has numerous formatting issues and should undergo extensive editing and proofreading before it can be considered for publication.

Detailed comments:
1.    Line 22: Missing quote after “green”.
2.    28: write “mixed effects” instead “mixed-effects”
3.    29: It reads like that variance components are estimated in a subsequent step. However, they are simultaneously estimated along with fixed effects. Also, note that random effects are integrated out in maximum likelihood estimation. Hence, random effects are predicted after the model has been estimated.
4.    105, 107: missing and unnecessary spaces after references
5.    The Excel figures look ugly. Please avoid this software for figure production in scientific publications.
6.    Figure 3: The outcome variable cannot take negative values, but the y-axis has a label with a negative value. This is awkward. Moreover, label the y-axis in millions to avoid writing many zeroes. Next, the upper-case labeling of the x-axis and y-axis must be avoided.
7.    Sect. 2.2. and throughout the paper: The authors do not care about the italic and non-italic fonts of mathematical symbols. Almost no symbol is consistently used in this article. Such careless work cannot be recommended for publication.
8.    Sect. 2.2.: Not all symbols in the MICE algorithm have been introduced. It might also be necessary to additionally refer to Raghunathan and van Buuren for this algorithm.
9.    254: I guess it must be Y_-1 instead of Y_1.
10.    349: What is Z_{(i),j}?
11.    Mixed effects models for (MICE) imputation have also been widely proposed in the literature. See the multilevel (mixed-effects) books of Goldstein or Wu. Moreover, consider the work of van Buuren, Carpenter and Quartagno, and Grund et al. At least, this literature should be cited.
12.    Figure 4: Do not use screenshots for figures.
13.    Figure 4: Why does “RUSSIA” appear in upper case? This is just another example of careless manuscript editing.
14.    The authors switch between “log”, “ln” and “Ln”. I prefer “log” in writing.
15.    471: Do not quote R and R packages.
16.    Figure 6: Remove the colored border. This is a journal article, not a figure in a magazine.
17.    507: The heading sounds awkward. Likely, it should read “MICE” instead of “NICE”.
18.    512 and others: Always write “Figure(s)” or “Equation(s)” in the manuscript. It must appear capitalized.
19.    Table 3 and others: If you have a look at published articles, you will recognize that it is standard practice to only report a relevant number of digits after the decimal (depending on the metric of the variables), not the number of digits that software reports as output.
20.    366: A more concise heading might be “model comparisons”.

Comments on the Quality of English Language

---

Reviewer 3 Report

Comments and Suggestions for Authors

Excellent article and very interesting. Can be interesting to include in your next article, the minning of REE in sediments of sea.

Round 2

Reviewer 2 Report

Comments and Suggestions for Authors

no further comments